# SPO: Sequential Monte Carlo Policy Optimisation

**Matthew V Macfarlane** *
University of Amsterdam
m.v.macfarlane@uva.nl

**Edan Toledo**
InstaDeep

**Donal Byrne**
InstaDeep

**Paul Duckworth**
InstaDeep

**Alexandre Laterre**
InstaDeep

## Abstract

Leveraging planning during learning and decision-making is central to the long-term development of intelligent agents. Recent works have successfully combined tree-based search methods and self-play learning mechanisms to this end. However, these methods typically face scaling challenges due to the sequential nature of their search. While practical engineering solutions can partly overcome this, they often result in a negative impact on performance. In this paper, we introduce SPO: Sequential Monte Carlo Policy Optimisation, a model-based reinforcement learning algorithm grounded within the Expectation Maximisation (EM) framework. We show that SPO provides robust policy improvement and efficient scaling properties. The sample-based search makes it directly applicable to both discrete and continuous action spaces without modifications. We demonstrate statistically significant improvements in performance relative to model-free and model-based baselines across both continuous and discrete environments. Furthermore, the parallel nature of SPO's search enables effective utilisation of hardware accelerators, yielding favourable scaling laws.

## 1 Introduction

The integration of reinforcement learning (RL) and neural-guided planning methods has recently achieved considerable success. Such methods effectively leverage planning during training via iterative imitation learning [7, 68, 18, 8]. Applying additional computation via planning generates improved policies which is then amortised [4] into a neural network policy. Using this policy as part of planning itself creates a powerful self-improvement cycle. They have demonstrated state of the art performance in applications ranging from chess [75] and matrix multiplication [24], to language modelling [61]. However, one of the most commonly-used search-based policy improvement operators, MCTS [16]: i) performs poorly on low budgets [33], ii) is inherently sequential limiting its scalability [51, 73], and iii) requires modifications to adapt to large or continuous action spaces [57, 41]. These limitations underscore the need for more scalable, efficient and generally applicable algorithms.

In this work, we introduce SPO: Sequential Monte Carlo Policy Optimisation, a model-based RL algorithm that utilises scalable sampled-based Sequential Monte Carlo planning for policy improvement. We formalise SPO as an approximate policy iteration algorithm, and show that it is formally grounded within the Expectation Maximisation (EM) framework.

---

*Work done during internship at InstaDeep

38th Conference on Neural Information Processing Systems (NeurIPS 2024).

SPO is unique by leveraging both breadth and depth of search in order to provide better estimates of the target distribution, derived via EM optimisation, compared to previous works. When viewed relative to popular algorithms it combines the breadth used in MPO [1] and the depth used in V-MPO [76], while enforcing KL constraints on updates to ensure stable policy improvement. SPO achieves strong performance across a range of benchmarks, outperforming AlphaZero, a leading model-based expert iteration algorithm. We also benchmark against a leading Sequential Monte Carlo (SMC) algorithm from Piché et al. [65] and show that leveraging posterior estimates from SMC for policy improvement is central to strong performance.

We demonstrate that i) SPO outperforms baselines across both high-dimensional continuous control and challenging discrete planning tasks without algorithmic modifications, and ii) the sampling-based approach is inherently parallelisable, enabling the use of hardware accelerators to improve training speed. This provides significant advantages over MCTS-based methods [16], whose sequential process can be particularly inefficient during training. 3) The explicit use of KL targeting leads to strong and stable policy improvement, reducing the need for a grid search over exploration hyperparameters in search. We find that SPO, empirically has strong scaling behaviours, with improved performance given additional search budget both during training and inference.

## 2 Related Work

There is an extensive history of prior works that frame control and RL as an inference problem [82, 45]. Expectation Maximization [19, 28, 60] is a popular approach to directly optimising the evidence lower bound (ELBO) of the probability of optimality of a given policy. A number of model free methods implement EM, including RWR [63], REPS [64], MPO [1], V-MPO [76], and AWAC [59] each with different approaches to the E-step and M-step (summarised in Furuta et al. [29]). Model based algorithms such as MD-GPS [58] and AlphaZero [74] can also be viewed under the EM framework using LQR [5] and MCTS [9, 16] during the E-step respectively. Grill et al. [33] show MCTS is a form of regularised policy optimisation equivalent to optimising the ELBO. Such model-based algorithms, that iterate between policy improvement using search, and projecting this improvement to the space of parameteriseable policies, are also often referred to as Expert iteration (ExIt) [74, 7] or Dual Policy Iteration [77]. Such approaches have also been applied to active inference [27] where MCTS has been used to estimate expected free energy, with this computation then amortised into a neural policy [25]. Our work can also be framed as ExIt, but differs by using Neural SMC for planning with an explicit KL regularisation on the improvement.

Sequential Monte Carlo (SMC) [31, 43, 52], also referred to as particle filters [66, 6], is an inference method often employed to sample from intractable distributions. Gu et al. [34] explore the parameterisation of the proposal in SMC using LSTMs [40], showing that posterior estimates from SMC can be used to train a parameterised proposal in non-control based tasks. Li et al. [49] demonstrate the same concept but using an MCMC sampler [30]. SMC can be applied to sample from distributions over trajectories in RL. For example, SMC-Learning [44] updates SMC weights using a Boltzmann exploration strategy, but unlike SPO it does not control policy improvement with KL regularisation or leverage neural approximators. Piché et al. [65] derive an SMC weight update to directly estimate the posterior over optimal trajectories. However this update requires the optimal value function, which they approximate using a SAC [37] value function. This approach requires high sample counts and does not leverage SMC posterior estimates for policy improvement, instead utilising a Gaussian distribution or SAC to train the proposal. CriticSMC [50] estimates the same posterior but utilises a soft action-value function to score particles [36], reducing the number of steps performed in the environment, enabling more efficient exploration and improved performance on lower SMC budgets compared with Piché et al. [65]. However their method is slower in wall clock time and uses a static proposal, therefore not performing iterative policy improvement. A useful properly of SMC search methods is that it is inherently parallelisable. Parallelising MCTS with a virtual loss has been explored, however this often leads to performance degradation [51], increasing exploration and leading to out-of-distribution states that are difficult to evaluate Dalal et al. [17].

## 3   Background

**Sequential decision-making** can be formalised under the Markov Decision Process (MDP) framework [67]. An MDP is a tuple $(\mathcal{S}, \mathcal{A}, \mathcal{T}, r, \gamma, \mu)$ where, $\mathcal{S}$ is the set of possible states, $\mathcal{A}$ is the set of actions, $\mathcal{T} : \mathcal{S} \times \mathcal{A} \to \mathcal{P}(\mathcal{S})$ is the state transition probability function, $r : \mathcal{S} \times \mathcal{A} \to \mathbb{R}$ is the reward function, $\gamma \in [0, 1]$ is the discount factor, and $\mu$ is the initial state distribution. An agent interacts with the MDP using a policy $\pi : \mathcal{S} \to \mathcal{P}(\mathcal{A})$, that associates to every state a distribution over actions. We quantify the quality of a policy by the expected discounted return, that the agent seeks to maximise $\pi^* \in \arg\max_{\pi \in \Pi} \mathbb{E}_\pi \left[ \sum_{t=0}^\infty \gamma^t r_t \right]$, where $r_t = r(s_t, a_t)$ is the reward received at time $t$ and $\Pi$ is the set of all realisable policies. The value function $V^\pi(s_t) = \mathbb{E}_\pi \left[ \sum_{t=t}^\infty \gamma^t r_t \mid s_t \right]$ maps a state $s_t$ to the expected discounted sum of future rewards when acting according to $\pi$. Similarly, the state-action value function $Q^\pi(s_t, a_t) = \mathbb{E}_\pi \left[ \sum_{t=t}^\infty \gamma^t r_t \mid s_t, a_t \right]$ maps a state $s_t$ to the expected discounted return, when taking the initial action $a_t$ and following $\pi$ thereafter.

**Control as inference** formulates the RL objective as an inference problem within a probabilistic graphical model [82, 42, 45]. For a horizon $T$, the distribution over trajectories $\tau = (s_0, a_0, s_1, a_1, ..., s_T, a_T)$ is given by $p(\tau) = \mu(s_0) \prod_{t=0}^T p(a_t) \mathcal{T}(s_{t+1} | s_t, a_t)$, which is a function of the initial state distribution, the transition dynamics, and an action prior. Note that this distribution is insufficient for solving control problems, because it has no notion of rewards. We therefore have to introduce an additional *optimality variable* into this model, which we will denote $\mathcal{O}_t$, that is defined such that $p(\mathcal{O}_t = 1 | \tau) \propto \exp(r_t)$. Therefore, a high reward at time $t$ means a high probability of having taken the optimal action at that point. We are then concerned with the target distribution $p(\tau | \mathcal{O}_{1:T})$, which is the distribution of trajectories given optimality at every step. We denote $\mathcal{O}_{1:T}$ as $\mathcal{O}$ going forward. The RL objective is then formulated as finding a policy to maximise $\log p_\pi(\mathcal{O} = 1) = \log \int \pi(\tau) p(\mathcal{O} = 1 | \tau) d\tau$, which intuitively can be thought of as maximising the distribution of optimality at all timesteps, given actions sampled according to $\pi$. To optimise this challenging objective, we can derive the evidence lower bound (ELBO) using an auxiliary distribution $q$ [60]:

$$\log p_\pi(\mathcal{O} = 1) = \log \int \pi(\tau) p(\mathcal{O} = 1 | \tau) d\tau = \log \int q(\tau) \frac{\pi(\tau) p(\mathcal{O} = 1 | \tau)}{q(\tau)} d\tau$$

$$\geq \int q(\tau) \left[ \log p(\mathcal{O} = 1 | \tau) + \log \frac{\pi(\tau)}{q(\tau)} \right] d\tau = \mathbb{E}_q \left[ \sum_t \frac{r_t}{\alpha} \right] - \mathrm{KL}(q(\tau) \| \pi(\tau)). \tag{1}$$

Since $p(\mathcal{O} = 1 | \tau) \propto \exp(\sum_t r_t)$ and $\alpha$ is a normalising constant, ensuring a valid probability distribution.

**Expectation Maximisation** (EM) [20] has been widely applied to solve the optimisation problem $\max_\pi \log p_\pi(\mathcal{O} = 1)$ [19, 63, 64, 1, 76]. After deriving the lower bound $\mathcal{J}(q, \pi)$ on the objective in eq. (1) using an auxiliary non-parametric distribution $q$, EM then performs a coordinate ascent, iterating between optimizing the bound with respect to $q$ (E-step) and with respect to the parametric policy $\pi$ (M-step). This generates a sequence of policy pairs $\{(\pi_0, q_0), (\pi_1, q_1), \ldots, (\pi_n, q_n)\}$ such that in each step $i$ in the EM sequence, $\mathcal{J}(q_{i+1}, \pi_{i+1}) \geq \mathcal{J}(q_i, \pi_i)$. Viewed through a traditional policy improvement lens, the E-step corresponds to a policy evaluation phase where we perform rollouts, generating states with their associated estimates for $q$ and value estimates. The M-step corresponds to a policy improvement phase where $\pi$ and $V$ are updated. This step can also be thought of as amortising the probabilistic inference computation performed in the E-step, into a neural network forward pass operation. [2]

**Sequential Monte Carlo** (SMC) methods [31] are designed to sample from an intractable distribution $p$ referred to as the *target distribution* by using a tractable proposal distribution $\beta$ (we use $\beta$ to prevent overloading of $q$). *Importance Sampling* does this by sampling from $\beta$ and weighting samples by $p(x)/\beta(x)$. The estimate is performed using a set of $N$ particles $\{x^{(n)}\}_{n=1}^N$, where $x^{(n)}$ represents a sample from the support that the target and proposal distributions are defined over, along with the associated importance weights $\{w^{(n)}\}_{n=1}^N$. Each particle uses a sample from the proposal distribution to improve the estimation of the target, increasing $N$ naturally improves the estimation of $p$ [10]. Once importance weights are calculated, the target is estimated as $\sum_{n=1}^N \bar{w}^{(n)} \delta_{x^{(n)}}(x)$,

---

[2]Many related RL algorithms can be understood by the different approaches to either the E-step or M-step, see appendix F.1 for a summary of EM approaches.

where $\bar{w}$ is the normalised importance sample weight and $\delta_{x^{(n)}}$ is the dirac measure located at $x^{(n)}$. *Sequential Importance Sampling* generalises this for sequential problems, where $x = (x_1, \ldots, x_T)$. In this case importance weights can be calculated iteratively according to:

$$w_t(x_{1:t}) = w_{t-1}(x_{1:t-1}) \cdot \frac{p(x_t|x_{1:t-1})}{\beta(x_t|x_{1:t-1})}. \tag{2}$$

Particle filtering methods can be affected by *weight degeneracy* [56]. This occurs when a few particles dominate the normalised importance weights, rendering the remaining particles negligible. As the variance of particle weights is guaranteed to increase over sequential updates [21], this phenomenon is unavoidable. When the majority of particles contribute little to the estimation of the target distribution, computational resources are wasted. *Sequential Importance Resampling* (SIR) [53] mitigates this problem by periodically resampling particles according to their current weights, subsequently resetting these weights.

## 4 SPO Method

In this section, we present a novel method that combines Sequential Monte Carlo (SMC) sampling with the Expectation Maximisation (EM) framework for policy iteration. We begin by formulating the objective function that we aim to optimise. We then outline our iterative approach to maximising this function, alternating between an expectation step (E-step) and a maximisation step (M-step). Within the E-step, we derive the analytical solution for optimising the objective with respect to the auxiliary distribution $q$. We then demonstrate how SMC can be employed to effectively estimate this *target distribution*. The M-step can be viewed as a projection of the non-parametric policy obtained in the E-step back onto the space of feasible policies. A comprehensive algorithmic outline of our proposed approach is provided in appendix D.2.

### 4.1 Objective

We add an additional assumption to the lower bound objective defined in eq. (1) and assume that the auxiliary distribution $q$, defined over trajectories $\tau$, can be decomposed into individual state dependent distributions, i.e. $q(\tau) = \mu(s_0) \prod_{t \geq 0} q(a_t|s_t) \mathcal{T}(s_{t+1}|s_t, a_t)$. We parameterise $\pi$ using $\theta$ which decomposes in the same way. This enables the objective to be written with respect to individual states instead of full trajectories. Multiplying by $\alpha$, the objective can then be written as follows:

$$\mathcal{J}(q, \pi_\theta) = \mathbb{E}_q \left[ \sum_{t=0}^{\infty} \gamma^t \left[ r_t - \alpha \mathrm{KL}(q(a|s_t) \parallel \pi(a|s_t, \theta)) \right] \right] + \log p(\theta). \tag{3}$$

Previous works have explored this formulation where $p(\theta)$ is a prior over the parameters of $\pi$ [38, 71, 1].

### 4.2 E-step

Within the expectation step of EM, we maximise eq. (3) with respect to $q$. As in previous work, instead of optimising for the rewards, we consider an objective written according to Q-values [1, 59, 54].

$$\max_q \mathbb{E}_{\mu(s)} \left[ \mathbb{E}_{q(a|s)} \left[ Q^q(s, a) \right] - \alpha \mathrm{KL}(q(\cdot|s) \parallel \pi(\cdot|s, \theta_i)) \right] \tag{4}$$

This aims to maximise the expected Q-value of a policy $q$, with the constraint that it doesn't move too far away from $\pi_i$. Framing optimisation with respect to Q-values is useful as it enables the use of function approximators to estimate $Q$.

Maximising this objective is difficult due to the dependence of both the expectation terms and Q-values on $q$. Following previous works, we fix the Q-values with respect to a fixed policy $\bar{\pi}$, resulting in a partial E-step optimisation [74, 1, 76]. We use the most recent estimate of $q$ as the fixed policy we perform maximisation with respect to, similar to AlphaZero's approach of acting according to the expert policy. The initial state distribution $\mu(s)$ is likewise fixed to be distributed according to $\mu_{\bar{\pi}}$. In practice, we use a FIFO replay buffer and sample from it to determine $\mu_{\bar{\pi}}$.

Equation (4) consists of balancing two objectives. To optimize this, we frame this as a constrained maximization problem, to avoid scaling problems, see appendix G.3 for further information. This objective limits the KL between $q$ and $\pi$ from exceeding a certain threshold:

$$\max_q \mathbb{E}_{\mu_{\bar{\pi}}(s)} \left[ \mathbb{E}_{q(a|s)} \left[ Q^{\bar{\pi}}(s,a) \right] \right]$$
$$\text{s.t. } \mathbb{E}_{\mu_{\bar{\pi}}(s)} \left[ \text{KL} \left( q(a|s) \| \pi(a|s,\theta_i) \right) \right] < \epsilon. \tag{5}$$

The following analytic solution to the constrained optimisation problem can then be derived using the Lagrangian multipliers method outlined in appendix G.1. Likewise $\eta^*$ is obtained by minimising the convex dual function eq. (7), and intuitively can be thought of as enforcing the KL constraint on $q$, preventing $q_i$ from moving to far from $\pi_i$:

$$q_i(a|s) \propto \pi(a|s,\theta_i) \exp \left( \frac{Q^{\bar{\pi}}(s,a) - V^{\bar{\pi}}(s)}{\eta^*} \right), \tag{6}$$

where $V^{\bar{\pi}}(s)$ is an action independent baseline. $Q(s,a) - V(s)$ is referred to as the advantage, where optimising for Q-values vs advantages is equivalent [59]. Optimising with advantages however, has demonstrated to practically outperform optimising Q-values directly [62, 76, 59]. This is also a natural update to perform as the *policy improvement theorem* [78] outlines that an update by policy iteration can be performed if at least one state-action pair has positive advantage and a non-zero probability of reaching such a state. Although we have a closed form solution for $q$, we do not have either the value function or action-value function in practice. In the next section we outline our approach to estimating this distribution.

**Regularised Policy Optimisation:** In our closed form solution to the optimisation eq. (6), we are required to solve for $\eta^*$ which ensures that the KL is constrained. This can be calculated by minimising the following objective [84], see appendix G.2 for derivation:

$$g(\eta) = \eta\varepsilon + \eta \int \mu(s) \log \left( \int \pi(a|s,\theta_i) \exp \left( \frac{A^{\bar{\pi}}(s,a)}{\eta} \right) da \right) ds. \tag{7}$$

Practically, we estimate eq. (7) using a sample-based estimator according to the distribution of states from the replay buffer and stored values for $\pi(a|s,\theta_i)$ and $A^{\bar{\pi}}(s,a)$.

### 4.2.1 Estimating Target Distribution

Building upon previous work using Sequential Monte Carlo for trajectory distribution estimation [44, 65], we propose a novel approach that integrates SMC within the Expectation Maximisation

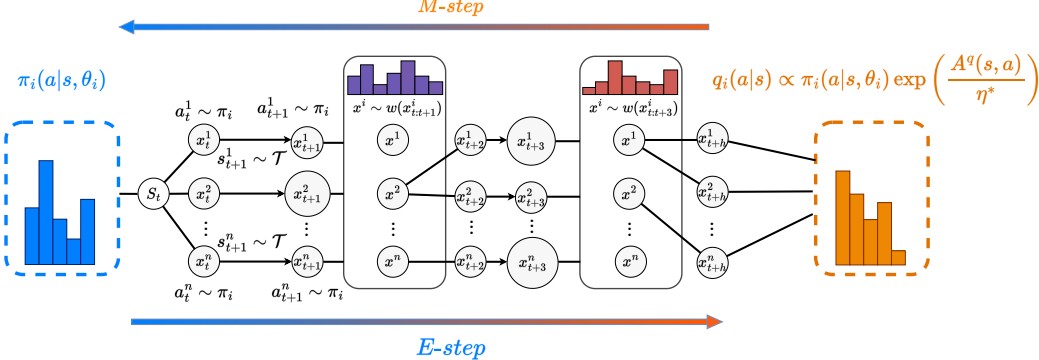

Figure 1: SPO search: $n$ rollouts, represented by particles $x^i, \ldots, x^n$, each of which represents an SMC trajectory sample, are performed in parallel according to $\pi_i$ (left to right). At each environment step, the weights of the particles are adjusted (indicated in the diagram by circle size). We show two resampling regions where particles are resampled, favouring those with higher weights, and their weights are reset. The target distribution is estimated from the initial actions of the surviving particles (rightmost particles). This target estimate, $q_i$, is then used to update $\pi$ in the M-step.

framework to estimate the target distribution defined in Equation 6. Our SMC-based method enables sampling from $q_i$ over trajectories of length $h$, incorporating multiple predicted future states and rewards for a more accurate distribution estimation. A key property of estimating the target using SMC is that it uses both breadth (we initialise multiple particles to calculate importance weights) but also depth (leveraging future states and rewards) to estimate the distribution. In contrast, MPO evaluates several actions per state, focusing on breadth without depth, while V-MPO calculates advantages using n-step returns from trajectory sequences, leveraging depth but only for a single action sample. Our method combines both aspects, enhancing the accuracy of target distribution estimation, crucial for both action selection and effective policy improvement updates in the M-step. An outline of the SMC algorithm is provided in algorithm 1, with a visual representation in fig. 1.

SMC estimates the target by maintaining a set of particles $\{x^{(n)}\}_{n=1}^N$ each of which maintains an importance weight for a particular sample. Given calculated importance weights SMC, estimates the target distribution according to $\hat{q}_i(\tau) = \sum_{n=1}^N \bar{w}^{(n)} \delta_{x_{(n)}}(\tau)$ where $\bar{w}^{(n)}$ is the normalised importance sample weight for a particular sample. We next outline the sequential importance sampling weight update needed to estimate eq. (6). Since we can sample from $\pi(a_t, s_t, \theta_i)$, we leverage this as our proposal distribution $\beta$. Given the target distribution $q$ can be decomposed into individual state dependent distributions, we can define $p_i(\tau_t|\tau_{1:t-1})$ and $\beta(\tau_t|\tau_{1:t-1})$ as:

$$p_i(\tau_t|\tau_{1:t-1}) \propto \mathcal{T}(s_{t+1}|s_t, a_t)\pi_i(a_t|s_t, \theta_i) \exp\left(\frac{\bar{A}_i(a_t, s_t)}{\eta_i^*}\right) \tag{8}$$

$$\beta(\tau_t|\tau_{1:t-1}) \propto \mathcal{T}_{\text{model}}(s_{t+1}|s_t, a_t)\pi_i(a_t|s_t, \theta_i) \tag{9}$$

leading to the following convenient SMC weight update according to eq. (2):

$$w(\tau_{1:t}) \propto w(\tau_{1:t-1}) \cdot \left(\frac{\mathcal{T}(s_t|s_{t-1}, a_{t-1})}{\mathcal{T}_{model}(s_t|s_{t-1}, a_{t-1})}\right) \cdot \frac{\exp\left(A^{\bar{\pi}}(a_t, s_t)/\eta^*\right) \cdot \pi(a_t|s_t, \theta_i)}{\pi(a_t|s_t, \theta_i)}, \tag{10}$$

where $Q^{\bar{\pi}}(a_t, s_t) - V^{\bar{\pi}}(s_t)$ is simplified to $A^{\bar{\pi}}(a_t, s_t)$, $\tau_{1:t}$ is a sequence of state, action pairs from timestep 1 to $t$, and $\mathcal{T}_{model}$ is the environment transition function of the planning model. Note that our work assumes the availability of a model that accurately represents the transition dynamics of the environment $\mathcal{T}$, and therefore simplify the update to $w(\tau_{1:t}) \propto w(\tau_{1:t-1}) \cdot \exp\left(A^{\bar{\pi}}(a_t, s_t)/\eta^*\right)$.

Algorithm 1 outlines the method for estimating eq. (6) over $h$ planning steps for $N$ particles (line 3) using the advantage based weight update (line 7) in eq. (10). Once $h$ steps in the environment model have been performed in parallel, we marginalise all but the first actions as a sample based estimate of $q_i$ (line 14).

The advantage function $A^{\bar{\pi}}$ is typically unknown, so we compute a 1-step estimate at each iteration using the value function and observed environment rewards $\hat{A}(s_t, a) = r_t + V^{\bar{\pi}}(s_{t+1}) - V^{\bar{\pi}}(s_t)$. Practically, we parameterise the value function $V$ using a neural network and train it using GAE [70] on true environment rollouts collected. After $h$ steps, the importance weights leverage $h$-steps of observed states and rewards during planning and corresponding advantage estimates. Compared to tree-based methods such as MCTS, SMC does not require maintaining the full tree in memory. Instead, after each planning step, it only needs to retain the initial action, current state, and current particle weight. It also does not require spending computation sending backward messages to update statistics of previous nodes everytime a new node is added to the tree.

---

**Algorithm 1** SMC $q$ target estimation (timestep $t$)

---

1: Initialize $\{s_t^{(n)} = s_t\}_{n=1}^N$
2: Set $\{w_t^{(n)} = 1\}_{n=1}^N$
3: **for** $i \in \{t+1, \ldots, t+h\}$ **do**
4:      $\{a_i^{(n)} \sim \pi(\cdot|s_i^{(n)}, \theta)\}_{n=1}^N$
5:      $\{s_{i+1}^{(n)} \sim \mathcal{T}_{model}(s_i^{(n)}, a_i^{(n)})\}_{n=1}^N$
6:      $\{r_i^{(n)} \sim r_{model}(s_i^{(n)}, a_i^{(n)})\}_{n=1}^N$
7:      $\{w_i^{(n)} = w_{i-1}^{(n)} \cdot \exp(\hat{A}(s_i^n, r_i^n, s_{i+1}^n)/\eta^*)\}_{n=1}^N$
8:      **if** $(i - t) \bmod p = 0$ **then**
9:          $\{x_{t:i}^{(n)}\}_{n=1}^N \sim \text{Mult}(n; w_i^{(1)}, .., w_i^{(N)})$
10:          $\{w_i^{(n)} = 1\}_{n=1}^N$
11:      **end if**
12: **end for**
13: $\{a_t^{(n)}\}$ as the set of first actions of $\{x_{t:t+h}^{(n)}\}_{n=1}^N$
14: $\hat{q}(a|s_t) = \sum_{n=1}^N \bar{w}^{(n)} \delta_{a_t^{(n)}}(a)$

---

**Resampling Adaptive Search:** To mitigate the issue of *weight degeneracy* [56], SPO conducts periodic resampling (lines 8-11). This involves generating a new set of particles from the existing set

by duplicating some trajectories and removing others, based on the current particle weights. This process enhances computational efficiency in estimating the target distribution. By resampling, we avoid updating weights for trajectories with low likelihood under the target, thereby reallocating computational resources to particles with high importance weights [22].

### 4.3 M-step

After completing the E-step (which generates $q_i$), we proceed with the M-step, which optimises eq. (3) with respect to $\pi$, parametrised by $\theta$. By eliminating terms that are independent of $\pi$, we optimise the following objective, corresponding to a maximum a posteriori estimation with respect to the distribution $q_i$:

$$\max_\theta \mathcal{J}(q_i, \pi_\theta) = \max_\theta \mathbb{E}_{\mu_{q_i}(s)} \left[ \mathbb{E}_{q_i(\cdot|s)} \left[ \log \pi(a|s, \theta) \right] \right] + \log p(\theta). \qquad (11)$$

This optimisation can be viewed as projecting the non-parametric policy $q_i$ back to the space of parametrisable policies $\Pi_\theta$, as performed in expert iteration style methods such as AlphaZero. $p(\theta)$ represents a prior over the parameter $\theta$. Previous works find that utilising a prior for $\theta$ to be close to the estimate from the previous iteration $\theta_i$ leads to stable training [1, 76]. Therefore we assume a gaussian prior over the current policy parameters, see appendix G.5 where we show utilising such a prior leads to the following constrained objective:

$$\max_\theta \mathbb{E}_{\mu_{q_i}(s)} \left[ \mathbb{E}_{q_i(a|s)} \left[ \log \pi(a|s, \theta) \right] \right]$$
$$\text{s.t.} \quad \mathbb{E}_{\mu_{q_i}(s)} \left[ \text{KL} \left( \pi(a|s, \theta_i), \pi(a|s, \theta) \right) \right] < \epsilon_m. \qquad (12)$$

### 4.4 Policy Improvement

Previous approaches to using SMC for RL either do not perform iterative policy improvement using SMC [65], or lack policy improvement constraints needed for iterative improvement [44]. Assuming that SMC provides perfect estimates of the analytic *target distribution* at each step $i$, Expectation Maximisation algorithm will guarantee that successive iterations will result in monotonic improvements in our lower bound objective, with details outlined in appendix G.4 [54].

**Proposition 1.** *Given a non-parametric variational distribution $q_i$ and a parametric policy $\pi_{\theta_i}$. Given $q_{i+1}$, the analytical solution to E-step optimisation eq. (3) , and $\pi_{\theta_{i+1}}$, the solution to maximisation problem in the M-step eq. (12) then the ELBO $\mathcal{J}$ is guaranteed to be monotonically increasing: $\mathcal{J}(q_{i+1}, \pi_{\theta_{i+1}}) \geq \mathcal{J}(q_i, \pi_{\theta_i})$.*

In practice we are unlikely to generate perfect estimates of the target through sample based inference and leave derivations regarding the impact of this estimation on overall convergence for future work. We also draw the connection between our EM optimisation method and Mirror Descent Guided Policy Search (MD-GPS) [58]. Our objective can be viewed as a specific instance of MD-GPS (see appendix G.6). Depending on whether dynamics are linear or not, optimising the EM objective can be viewed either as exact or approximate mirror descent [11]. Monotonic improvement guarantees in MD-GPS follow from those of mirror descent.

## 5 Experiments

In this section, we focus on three main areas of analysis. First, we demonstrate the improved performance of SPO in terms of episode returns, relative to both model-free and model-based algorithms. We conduct evaluations across a suite of common environments for both continuous control and discrete action spaces. Secondly, we examine the scaling behaviour of SPO during training, showing that asymptotic performance scales with particle count and depth. Finally, we explore the performance-to-speed trade-off at test time by comparing SPO directly to AlphaZero as the search budget increases.

### 5.1 Experimental Setup

In order to ensure the robustness of our conclusions, we follow the evaluation methodology proposed by Agarwal et al. [3], see appendix H for further details. This evaluation methodology groups performance across tasks within an environment suite, enabling clearer conclusions over the significance of

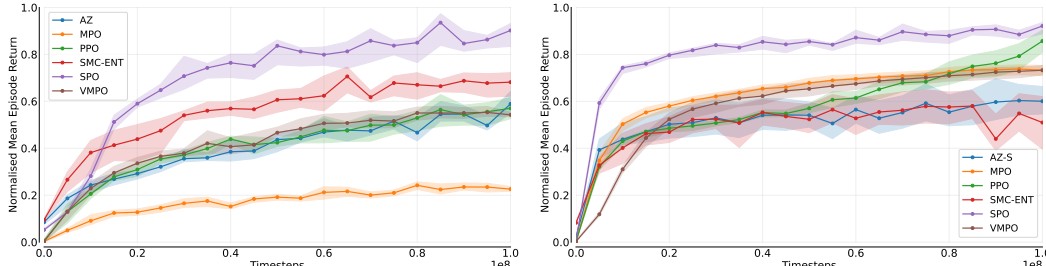

**(a) Discrete:** Rubik's Cube 7 and Boxoban Hard. **(b) Continuous:** Ant, HalfCheetah, Humanoid.

Figure 2: Learning curves for discrete and continuous environments. The Y-axis represents the interquartile mean of min-max normalised scores, with shaded regions indicating 95% confidence intervals, across 5 random seeds.

learning algorithms as a whole. We include individual results in appendix C, along with additional analysis measuring the statistical significance of our results.

**Environments:** For continuous control we evaluate on the Brax [26] benchmark environments of: Ant, HalfCheetah, and Humanoid. For discrete environments, we evaluate on Boxoban [35] (a specific instance of Sokoban), commonly used to assess planning methods, and Rubik's Cube, a sparse reward environment with a large combinatorial state-action space. See appendix A for further details regarding environments.

**Baselines:** Our model-free baselines include PPO [72], MPO [1] and V-MPO [76] for both continuous and discrete environments. For model-based algorithms, we compare performance to AlphaZero (including search improvements from MuZero [68]) and an SMC method introduced by Piché et al. [65] [3] (which we refer to as SMC-ENT due to its use of maximum entropy RL to train the proposal and value function used within SMC). Our AlphaZero benchmark follows the official open-source implementation[4]. For continuous environments we baseline our results to Sampled MuZero [41], a modern extension to MuZero for large and/or continuous action spaces. We utilise a true environment model, aligning our implementation of Sampled MuZero more closely with AlphaZero. Our core experiments configure SPO with 16 particles and a horizon of 4, and AlphaZero with 64 simulations to equalise search budgets. For remaining parameters, see appendix E.1 and appendix D.3. Each environment and algorithm were evaluated with five random seeds. [5]

### 5.2 Results

**Policy Improvement:** In fig. 2 we show empirical evidence that SPO outperforms all baseline methods across both discrete (left) and continuous (right) benchmarks for policy improvement. This conclusion also holds for the per-environment results reported in appendix C. SMC-ENT [65] is a strong SMC based algorithm, however, SPO outperforms it for both discrete and continuous environments providing strong evidence of the direct benefit of using SMC posterior estimates for policy improvement. We also highlight the importance regularising policy optimisation, which is practically achieved by solving for $\eta^*$ eq. (7), and is re-estimated at each iteration. In appendix B we ablate the impact of this adaptive method by comparing varying fixed temperature schedules. While a good temperature can be found through expensive grid search for each new environment we find the adaptive temperature ensures stable updates leading to strong learning performance across all environments, with minimal overhead to compute $\eta^*$.

We find that SPO performance is statistically significant compared to AlphaZero across the evaluated environments, see appendix C for detailed results. The substantial variation in AlphaZero performance across these environments underscores the difficulty in tuning it for diverse settings. For instance, while AlphaZero performs well on Boxoban and HalfCheetah, its performance drops significantly on other discrete and continuous problems. This inconsistency poses a major challenge

---

[3] We use MDQN [83], and SAC [37] for discrete and continuous environments, respectively. Note that in the discrete case, SMC-ENT using SAC was very unstable, so we chose MDQN as a better alternative.

[4] Open-source implementation available at mctx.

[5] Inference code and checkpoints are available at `https://github.com/instadeepai/spo`

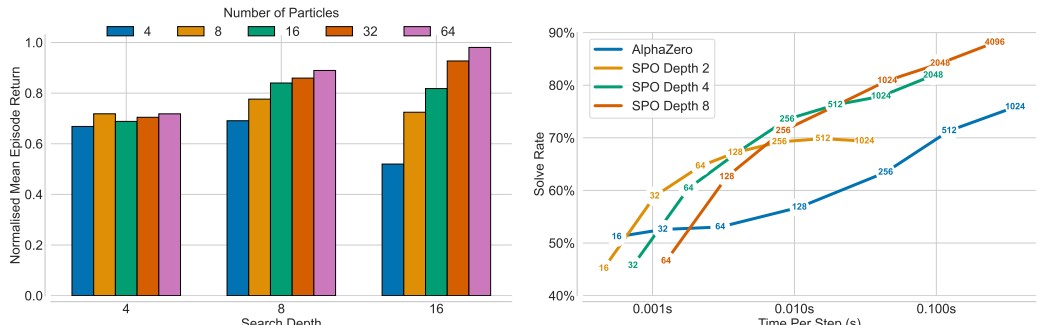

Figure 3: (left) Scaling: Mean normalised performance across all continuous environments on $10^8$ environment steps, varying particle numbers $N$ and horizon $h$ for SPO during training. (right) Wall Clock Time Comparison: Performance on Rubik's cube plotted against wall-clock time for AlphaZero and 3 versions of SPO (varying by SMC search depth), with total search budget labeled at each point.

for its applicability to real-world problems. In contrast, SPO performs consistently well across all environments (both discrete and continuous), highlighting its general applicability and robustness to various problem settings.

## 5.3   Scaling SPO during training

In fig. 3 (left) we investigate how SPO performance is impacted by scaling particle counts $N$ and horizon length $h$ during training, both of which we find improve the estimation of the *target distribution*. Our results show that scaling both the particle count and the horizon leads to improvements in the asymptotic performance of SPO. It also suggests that both variables should be scaled together for maximum benefit as having a long horizon with low particle counts can actually negatively impact performance. Secondly, we highlight that while for our main results we use a total search budget (particles $\times$ depth) of $64$ for policy improvement, our scaling results show that competitive results can be achieved by reducing this budget by a factor of four, which demonstrates competitive performance at low compute budgets.

## 5.4   Scaling SPO at test time

We can scale particle counts and horizon during training to improve target distribution estimates for the M-step. However, this scaling can also be performed after training when $\pi_{\theta_i}$ is fixed, enhancing performance, since actions are sampled in the environment according to the improved policy $q$. This equates to additional E-step optimization, which can enhance performance due to the representational constraints of the space of parameterised policies. In fig. 3 (right) we demonstrate how the performance of both SPO and AlphaZero search scales with additional search budget at test time using the same high performing checkpoint. We plot the time taken for a single actor step (measured on a TPUv3-8) against solve rate for the Rubik's Cube problem with time on a logarithmic axis. Specifically, we evaluate on 1280 episodes for cubes ten scrambles away from solved. While we recognise that such analysis can be difficult to perform due to implementation discrepancies, we used the official JAX implementation of AlphaZero (MCTX) within our codebase and compare this to SPO. Additionally, we exclusively measure inference thus no training implementation details affect our measurements.

This provides evidence that AlphaZero has worse scaling when compared to SPO on horizons four and eight, noting that for horizon of two, SPO performance converges early, as low depths can act as a bottleneck for further performance improvements. This highlights the benefits of the SPO parallelism. This chart also shows how for different compute preferences at test time, a different horizon length is preferable. For limited compute, low horizon lengths and relatively higher particle counts provide the best performance, but as compute availability increases, the gains from increasing particle counts decrease and instead horizon length $h$ should be increased.

Lastly, the plot illustrates the increase in available search budget, by using SPO, given a time restriction rather than a compute restriction. For example, SPO can use 4 times more search budget, when compared to AlphaZero, given a requirement of 0.1 second per step.

# 6  Conclusion

Planning-based policy improvement operators have proven to be powerful methods to enhance the learning of policies for complex environments when compared to model-free methods. Despite the success of these methods, they are still limited in their usefulness, due to the requirement of high planning budgets and the need for algorithmic modifications to adapt to large or continuous action spaces. In this paper, we introduce a modern implementation of Sequential Monte Carlo planning as a policy improvement operator, within the Expectation Maximisation framework. This results in a general training methodology with the ability to scale in both discrete and continuous environments.

Our work provides several key contributions. First, we show that SPO is a powerful policy improvement operator, outperforming our model-based and model-free baselines. Additionally, SPO is shown to be competitive across both continuous and discrete domains, without additional environment-specific alterations. This illustrates the effectiveness of SPO as a generic and robust approach, in contrast to prior methods that require domain-specific enhancements. Furthermore, the parallelisable nature of SPO results in efficient scaling behaviour of search. This allows SPO to achieve a significant boost in performance at inference time by scaling the search budget. Finally, we demonstrate that scaling SPO results in faster wall-clock inference time compared to previous work utilising tree-based search methods. The presented work culminates in a versatile, neural-guided, sample-based planning method that demonstrates superior performance and scalability over baselines.

**Limitations & Future Work:** This work demonstrates the efficacy of combining probabilistic inference with reinforcement learning and amortisation. While our work considers a relatively standard form of Sequential Monte Carlo, future work could investigate making improvements to this inference method, in order to further improve the estimate of the target distribution, while maintaining the scalability benefits demonstrated. We also draw the connection to Active Inference [27] which has been explored in the context of agent behaviour in complex environments, where Monte Carlo Tree Search has been used to scale previous methods along with deep learning [25]. Leveraging Sequential Monte Carlo in such methods along with amortisation could be a promising direction of research to further scale such methods. Our work exclusively uses exact world models of the environment in order to isolate the effects of various planning methods on performance. Extending our research to include a learned world model would broaden the applicability of SPO to more complex problems that may not have a perfect simulator and are therefore unsuitable for direct planning. Additionally, we apply Sequential Monte Carlo (SMC) only in deterministic settings. Adapting our approach for stochastic environments presents challenges. Specifically, the importance weight update in SMC can lead to the selection of dynamics most advantageous to the agent, potentially fostering risk-seeking behaviour in stochastic settings. However, mitigation strategies exist, as discussed in Levine [45].

## Acknowledgments and Disclosure of Funding

The authors would like to thank Teodora Pandeva for detailed discussions on SPO theory and for detailed revisions of the manuscript. Clément Bonnet for brainstorming discussions and feedback and David Kuric for revisions of the final manuscript. We also thank members of the InstaDeep team for their support in the preparation of the final manuscript. Lastly we thank the anonymous reviewers for comments and helpful discussions that helped improve the final version of the paper.

Research was supported with Cloud TPUs from Google's TPU Research Cloud (TRC). Matthew Macfarlane is supported by the LIFT-project 019.011, which is partly financed by the Dutch Research Council (NWO).

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

# Appendix

## Table of Contents

# A Environments

## A.1 Sokoban

### A.1.1 Overview

Table 1: Summary of Boxoban dataset levels

| Difficulty Level | Dataset Size |
| --- | --- |
| Unfiltered-Train | 900k |
| Unfiltered-Validation | 100k |
| Unfiltered-Test | 1k |
| Medium | 450k |
| Hard | 50k |

We use the specific instance of Sokoban outlined in Guez et al. [35] illustrated in Figure 4 with the codebase available at `https://github.com/instadeepai/jumanji`. The datasets employed in this study are publicly accessible at `https://github.com/google-deepmind/boxoban-levels`. These datasets are split into different levels of difficulty, which are categorised in Table 1.

In this research, we always train on the *Unfiltered-Train* dataset. Evaluations are performed on the *Hard* dataset, important for differentiating the strongest algorithms.

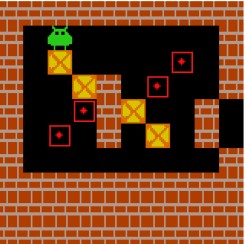

Figure 4: Example of a Boxoban Problem

### A.1.2 Network Architecture

Observations are represented as an array of size (10,10) where each entry is an integer representing the state of a cell. We used a deep ResNet [39] architecture for the torso network to embed the observation. We define a block as consisting of the following layers: [CNN, ResNet, ResNet]. Four such blocks are stacked, each characterized by specific parameters:

- Output Channels: (256, 256, 512, 512)
- Kernel Sizes: (3, 3, 3, 3)
- Strides: (1, 1, 1, 1)

Additionally, the architecture includes two output heads (policy and value), each comprising two layers of size 128 with ReLU activations. The output heads share the same torso network. We use the same architecture for all algorithms.

## A.2 Rubik's Cube

### A.2.1 Overview

For our Rubik's Cube experiments, we utilize the implementation in Bonnet et al. [12] available at `https://github.com/instadeepai/jumanji`. Rubik's cube problems can be be made progressively difficult by scaling the number of random actions performed in a solution state to generate problem instances. We always perform training on a uniform distribution sampled from the range [3,7] of scrambled states, followed by exclusively evaluating on 7 scrambles. The observation is represented using an array of size (6,3,3). The action space is represented using an array of size (6,3) corresponding to each face and the depth at which it can be rotated.

### A.2.2 Network Architecture

We utilise an embedding layer to first embed the face representations to size (6,3,3,4). The embedding is flattened and we use a torso layer consisting of a two layer residual network with layer size 512 and ReLU activations. Additionally, the architecture includes two output heads (policy and value), each comprising of a single layer of size 512 with ReLU activations. We use the same architecture for all algorithms.

## A.3 Brax

### A.3.1 Overview

For experimentation on continuous settings, we make use of Brax [26]. Brax is a library for rigid body simulation much like the popular MuJoCo setting [79] simulator. However, Brax is written entirely in JAX [14] to fully exploit the parallel processing capabilities of accelerators like GPUs and TPUs.

It is important to note, that at the time of writing, there are 4 different physics back-ends that can be used. These back-ends have differing levels of computational complexity and the results between them are not comparable. The results generated in this paper utilise the Spring backend.

Table 2: Observation and action space for Brax environments

| Environment | Observation Size | Action Size |
|---|---|---|
| Halfcheetah | 18 | 6 |
| Ant | 27 | 8 |
| Humanoid | 376 | 17 |

Table 2 contains the observation and action specifications of the scenarios used for our experiments. We specify the dimension size of the observation and action vectors.

### A.3.2 Network Architecture

In practice, we found the smaller networks used in the original Brax publication to limit overall performance. We instead use a 4 layer feed forward network[6] to represent both the value and policy network. Aside from the output layers, all layers are of size 256 and use SiLU non-linearities. We use the same architecture for all algorithms. Unlike Rubik's Cube and Sokoban, we do not use a shared torso to learn embeddings.

---

[6]except for SMC-ENTR,for which we use 3 to stabilise performance.

# B Ablations

## B.1 E-step KL constraint

The KL constraint for the E-step within the Expectation Maximisation framework is an important component of ensuring policy improvement, due to the KL term in the ELBO. We investigate the impact of the adaptive temperature for both discrete and continuous environments. We compare SPO with an adaptive temperature updated every iteration to a variety of fixed temperatures.

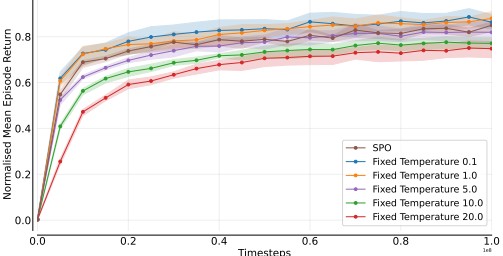
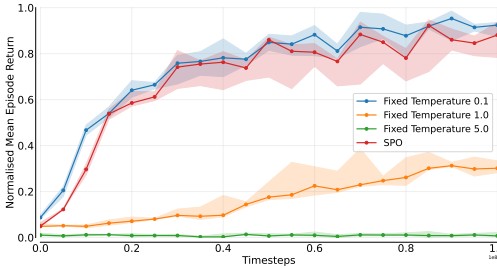

Figure 5: **Brax**                    Figure 6: **Sokoban and Rubik's Cube**

This ablation shows that SPO with an adaptive temperature is among the top performing hyperparameter settings across all environments. However we also note that it is possible to tune a temperature that works well when considering a wide range of temperatures. This is consistent with previous results in Peng et al. [62] that also find practically for specific problems a fixed temperature can be used. Of course in practice having an algorithm that can learn this parameter itself is practically beneficial, removing the need for costly hyperparameter tuning, since the appropriate temperature is likely problem dependant.

Subsequently, we evaluated whether the partial optimisation of the temperature parameter $\eta$ effectively maintained the desired KL divergence constraint and how different values of this constraint affected performance.

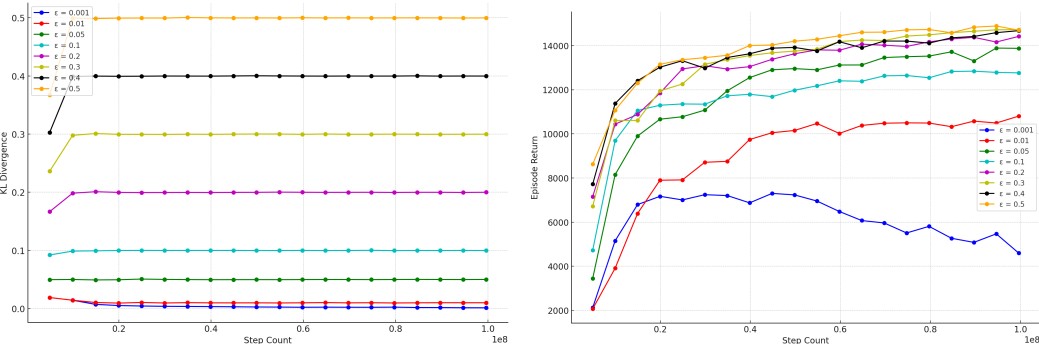

Figure 7: (a) The estimated KL divergence between the prior policy $\pi$ and the target policy $q$ generated by SMC for different values of $\epsilon$ during training on the Brax Ant task. (b) Evaluation performance during training for different values of $\epsilon$.

As shown in Figure 7(a), the KL divergence constraint is tightly maintained throughout training for different values of $\epsilon$. Figure 7(b) demonstrates that, for the Ant task, allowing a larger KL divergence between the prior and target policies ($q$ and $\pi$, respectively) leads to improved performance.

## B.2 E-step Q-value Optimisation

For our analytic solution to the optimisation problem in the E-step we choose to add a value baseline such that our target distribution re-weights with respect to Advantages instead of Q-values. In practice there are no restrictions on utilising Q-values instead for the importance weight update of SMC eq. (10). Below we show results on the continuous environments comparing SPO using advantages to SPO using Q-values.

We see in Figure 8, that the use of Q values produces a reduction in performance that is relatively constant throughout training. It is possible that this reduction is due to different hyperparameters being required to effectively utilise the Q-value distribution but in order to accurately judge the effect, we kept hyperparameters constant

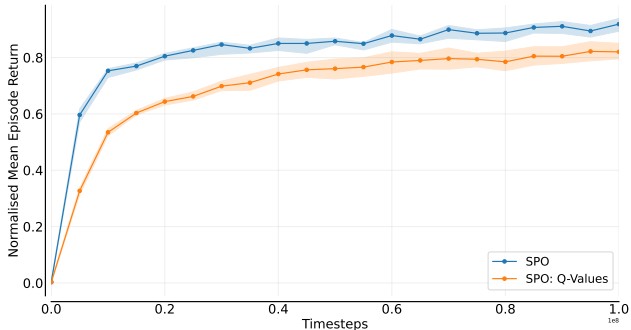

Figure 8: Q-value ablation on Brax tasks

across the runs. Secondly we note this is consistent with existing results for EM algorithms, demonstrating that Advantages outperform Q-values Peng et al. [62], Song et al. [76], Nair et al. [59].

## B.3 SMC Target Estimation Validation

Estimating the target distribution accurately is important for policy improvement in SPO. Therefore, we aim to validate the ability of SMC search to better approximate the true target distribution than a simple 1-step function evaluation as is done in algorithms such as MPO [1]. To do this, while the true target distribution is not directly accessible, we approximate it using an unbiased Monte Carlo oracle. We use a large computational budget of 1280 rollouts, to the end of the episode, for every state and every action. We evaluate the impact of varying planning horizons (depth) and particle counts on the KL divergence between the SMC-estimated target policy and the Monte Carlo oracle in the Sokoban environment.

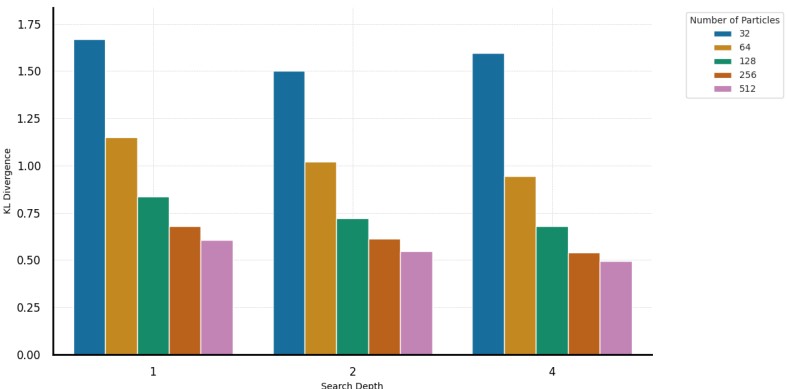

Figure 9: Comparison of the KL divergence to large Monte Carlo simulation of target policy for different planning horizons and particle counts for Sokoban

As shown in Figure 9, increasing the number of particles and the planning depth reduces the KL divergence to the oracle, indicating improved estimation of the target distribution. This aligns with SMC theory [22], which suggests that more particles and deeper planning lead to better approximations. These results highlight the importance of leveraging both breadth (more particles) and depth (longer planning horizons) in SMC for accurate target estimation.

# C    Expanded Results

We provide more detailed analysis of the results presented in the main body of the paper, including the final point aggregate performance of algorithms, the probability of improvement plots and the performance profiles. All plots are generated according to the methodology outlined by Agarwal et al. [3] and Gorsane et al. [32] by performing the final evaluation using parameters that achieved the best performance throughout intermediate evaluation runs during training.

The point aggregation plots show the rankings of normalised episode returns of each algorithm when different aggregation metrics are used. The probability of improvement plots illustrate the likelihood that algorithm $X$ outperforms algorithm $Y$ on a randomly selected task. It is important to note that a statistically significant result from this metric is a probability of improvement greater than 0.5 where the confidence intervals (CIs) do not contain 0.5. The metric utilises the Mann-Whitney U-statistic [55]. See [3] for further details. The performance profiles illustrate the fraction of the runs over all training environments from each algorithm that achieved scores higher than a specific value (given on the X-axis). Functionally, this serves the same purpose as comparing average episodic returns for each algorithm in a tabular form but in a simpler format. Additionally, if an algorithm's curve is strictly greater than or equal to another curve, this indicates "stochastic dominance" [47, 23].

## C.1    Detailed Summary Results

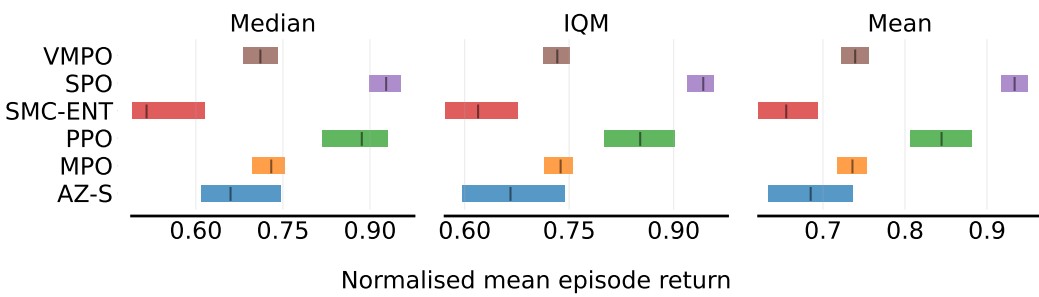

Figure 10: Aggregate point metrics for Brax suite. 95% confidence intervals generated from stratified bootstrapping across tasks and seeds are reported.

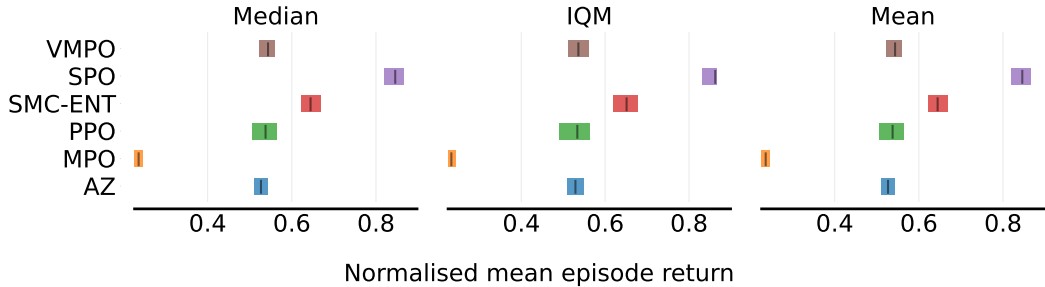

Figure 11: Aggregate point metrics for Sokoban and Rubik's Cube. 95% confidence intervals generated from stratified bootstrapping across tasks and seeds are reported.

In Figures 10 and 11, we present the absolute metrics following the recommendations by Agarwal et al. [3], Colas et al. [15] and Gorsane et al. [32]. These metrics evaluate the best set of network parameters identified during 20 evenly spaced training evaluation intervals. The absolute evaluations measure performance across 10 times the number of episodes periodically assessed during training, resulting in a total of 1280 episodes evaluated. We report point estimates and their 95% confidence intervals, which were calculated using stratified bootstrapping. Agarwal et al. [3] advocate for the Interquartile Mean (IQM) as a more robust and valid point estimate metric. Our results indicate that SPO surpasses all other algorithms in every point estimate. Notably, the IQM and mean point estimates demonstrate statistical significance.

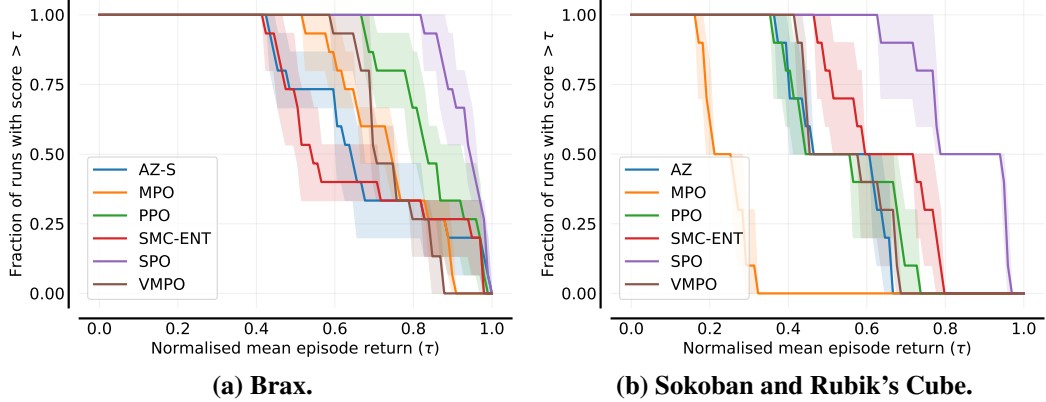

(a) Brax.  (b) Sokoban and Rubik's Cube.

Figure 12: Performance profiles. The Y-axis represents the fraction of runs that achieved greater than a specific normalised score represented on the x-axis, with shaded regions indicating 95% confidence intervals generated from stratified bootstrapping across both tasks and random seeds.

In Figure 12, we present the performance profiles which visually illustrates the full distribution of scores across all tasks and seeds for each algorithm. We see that SPO has a higher lower bound on performance, in addition to upper bound, indicating lower variance across tasks and seeds. Additionally, we see that SPO is strictly above all other algorithms indicating that SPO is stochastically dominant[7] to all baselines.

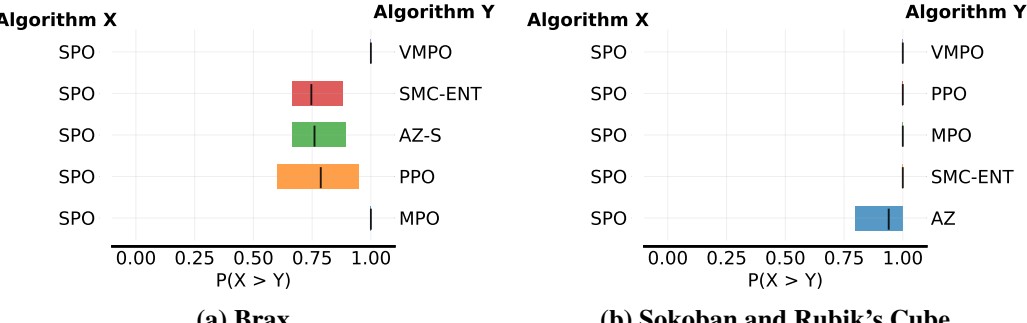

(a) Brax.  (b) Sokoban and Rubik's Cube.

Figure 13: Probability of Improvement.

Lastly, we present the probability of improvement plots in Figure 13. We see that SPO has a high probability of improvement compared to all baselines. Additionally, all probabilities are larger than 0.5 and have their CIs above 0.5 thus indicating statistical significance as specified by Agarwal et al. [3]. Furthermore, we see all CIs have upper bounds greater than 0.75, thereby indicating that these results are statistically meaningful as specified by Bouthillier et al. [13].

## C.2  Hardware

Training was performed using a mixture of Google v4-8 and v3-8 TPUs. Each experiment was run using a single TPU and only v3-8 TPUs were used to compare wall-clock time.

---

[7]A random variable $X$ is termed stochastically dominant over another random variable $Y$ if $P(X > \tau) \geq P(Y > \tau)$ for all $\tau$, and for some $\tau$, $P(X > \tau) > P(Y > \tau)$.

## C.3 Individual Environment Results

All individual task results are presented in fig. 14. Specifically, we present the IQM of returns achieved during the evaluation intervals throughout training.

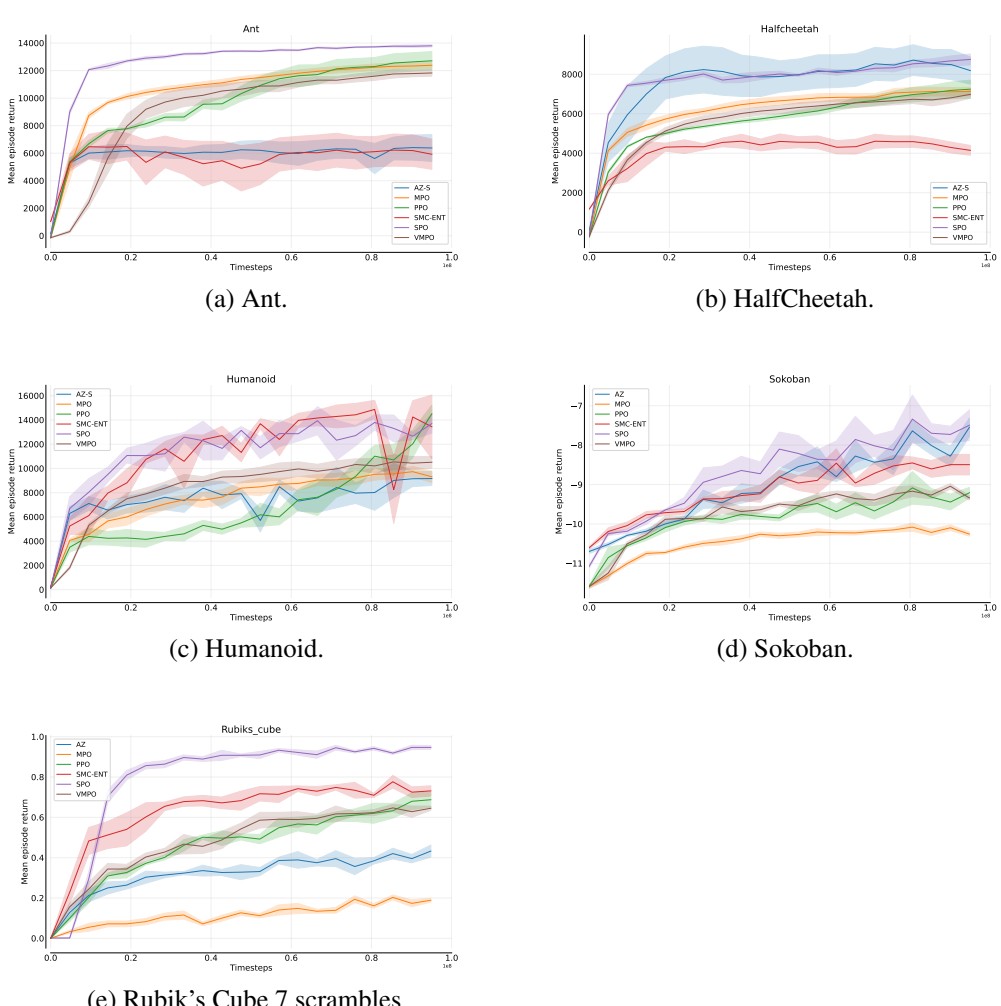

(a) Ant.

(b) HalfCheetah.

(c) Humanoid.

(d) Sokoban.

(e) Rubik's Cube 7 scrambles

Figure 14: Performance across different environments.

# D  SPO

Below we outline the practical policy- based losses used within SPO,

$$\mathcal{L}_\pi(\theta) = - \sum_{s,a \sim \mathcal{D}} q_i(a|s) \log \pi_{\theta_i}(a|s)$$

$$g(\eta) = \eta\varepsilon + \eta \int \mu(s) \log \left( \int \pi(a|s,\theta_i) \exp \left( \frac{A^{\bar{\pi}}(s,a)}{\eta} \right) da \right) ds.$$

$$\mathcal{L}_\alpha(\theta, \alpha) = \alpha \left( \epsilon_\alpha - \mathbb{E}_{s \sim p(s)} \left[ \mathrm{sg} \left[ \mathcal{D}_{\mathrm{KL}}(\pi_{\theta_{\mathrm{old}}} \| \pi_\theta)] \right] \right) \right.$$

$$\mathcal{L}_{KL}(\theta, \alpha) = \mathrm{sg}\left[\alpha\right] \mathbb{E}_{s \sim p(s)} \left[ \mathcal{D}_{\mathrm{KL}}(\pi_{\theta_{\mathrm{old}}} \| \pi_\theta) \right]$$

## D.1  Temperature Loss

The temperature loss is calculated using advantages collected during SPO search with rollouts performed according to policy $\pi$. If resampling is utilised, after a resample the distribution over trajectories shifts towards the target distribution away from $\pi$. Therefore in order to calculate the loss $g(\eta)$, we only utilise the advantages up to the first resampling period with rollouts performed according to $\pi$.

## D.2  Approximate Policy Iteration Algorithm

Below we outline the overall SPO algorithm with the main loop split into the E-step and M-step.

---

**Algorithm 2** SPO Algorithm

---

1: $\mathcal{B} \leftarrow \emptyset$
2: **Initialize the policy, value function, temperature, and alpha:**
3:     Initialize $\theta_0, \phi_0, \eta_0, \alpha_0$
4: **for** iteration $k = 0, 1, 2, \ldots, $ num_iterations **do**
5:     **Expectation Step (E-step):**
6:     **for** agent $m = 1, 2, \ldots, M$ **in parallel do**
7:         Initialize state $s_0$
8:         **for** timestep $i = 1, 2, \ldots, $ rollout_length **do**
9:             $\bar{a} = \{a^{(n)}\}_{n=1}^N \sim \hat{q}_{SMC}(s_i, \eta_k, \theta_k, \phi_k)$         ▷ sample-based estimate of $q_i$
10:             $a \sim \{a^{(n)}\}_{n=1}^N$         ▷ Sample an action to execute in the environment
11:             $s_{i+1} \sim \mathcal{T}(s_i, a)$         ▷ Execute action $a$, observe next state $s_{i+1}$
12:             $r_i \sim r(s_i, a)$         ▷ Observe reward $r_i$
13:             Store $(s_i, r_i, \bar{a})$ in $\mathcal{B}$
14:         **end for**
15:     **end for**
16:     **Maximization Step (M-step):**
17:     **for** batch $b = 1, 2, \ldots, $ num_batches **do**
18:         **Sample batch from replay buffer:**
19:             Sample batch of size $N$ from replay buffer $(s, r, \bar{a}) \sim \mathcal{B}$
20:         **Update value function using GAE targets:**
21:             $\phi_{k+1} \leftarrow \arg\min_\phi \mathbb{E}_{s \sim \mathcal{B}} \left[ (V_{\mathrm{GAE}}(s, \phi') - V(s, \phi))^2 \right]$
22:         **Update parameterized policy using sample-based estimate of** $q$**:**
23:             $\theta_{k+1}, \alpha_{k+1} \leftarrow \arg\min_\theta \mathbb{E}_{s \sim \mathcal{B}} \left[ \mathcal{L}_\alpha(\theta, \alpha) + \mathcal{L}_{KL}(\theta, \alpha) \right]$
24:         **Update dual temperature:**
25:             $\eta_{k+1} \leftarrow \arg\min_\eta \mathbb{E}_{s \sim \mathcal{B}} \left[ g(\eta) \right]$
26:         $\phi' \leftarrow \mathrm{polyak}(\phi', \phi_{k+1})$
27:     **end for**
28: **end for**

---

## D.3 Hyperparameters

Table 3: SPO Hyperparameters for Continuous and Discrete Environments

| Parameter | Continuous Environments | Discrete Environments |
|---|---|---|
| Actor & Critic Learning Rate | $3e^{-4}$ | $3e^{-4}$ |
| Dual Learning Rate | $1e^{-3}$ | $3e^{-4}$ |
| Discount Factor | 0.99 | 0.99 |
| GAE Lambda | 0.95 | 0.95 |
| Replay Buffer Size | $6.5e^4$ | $6.5e^4$ |
| Batch Size | 32 | 64 |
| Batch Sequence Length | 32 | 17 |
| Max Grad Norm | 0.5 | 0.5 |
| Number of Epochs | 128 | 16 |
| Number of Envs | 1024 | 768 |
| Rollout Length | 32 | 21 |
| $\tau$ (Target Smoothing) | $5e^{-3}$ | $5e^{-3}$ |
| Number of Particles | 16 | 16 |
| Search Horizon | 4 | 4 |
| Resample Period | 4 | 4 |
| Initial $\eta$ | 10 | 0.5 |
| Initial $\alpha$ | - | 0.5 |
| Initial $\alpha_\mu$ | 10 | - |
| Initial $\alpha_\Sigma$ | 500 | - |
| $\epsilon_\eta$ | 0.2 | 0.5 |
| $\epsilon_\alpha$ | - | $1e^{-3}$ |
| $\epsilon_{\alpha_\mu}$ | $5e^{-2}$ | - |
| $\epsilon_{\alpha_\Sigma}$ | $5e^{-4}$ | - |
| Dirichlet Alpha | - | 0.03 |
| Root Exploration Weight | - | 0.25 |

### D.4 Intuition for Temperature and Exploration:

The temperature parameter $\eta$ plays a crucial role in balancing exploration and exploitation during the search process in SPO. In this work, it is derived via first choosing the desired KL divergence $\epsilon$ between the target policy $q_i$ generated by search and the current policy $\pi_i$. If the temperature is too low (i.e., $\eta$ is small), the exponential weighting $\exp\left(A^{\bar{\pi}}(s,a)/\eta\right)$ becomes sharply peaked around the highest-advantage actions. This causes the importance weights to concentrate on a few particles, and during resampling, the particles can collapse onto a single root action. Such collapse reduces diversity in the search and renders the rest of the planning process ineffective. Conversely, if the temperature is too high (i.e., $\eta$ is large), the weighting becomes flatter, leading to excessive exploration. While exploration is necessary, too much can prevent the search from effectively focusing on promising paths. This work shows that deriving this value via a target KL leads to stable and strong performance across different environments.

## E  Baselines

For the baseline implementations, we expanded and adapted the existing implementations from Stoix[8] [80]. All implementations were conducted using JAX [14], with off-policy algorithms leveraging Flashbox [81].

### E.1  Hyperparameters

#### E.1.1  Continuous Control

Table 4: Hyperparameters for PPO and Sampled AlphaZero

| Parameter | PPO | Sampled AlphaZero |
|---|---|---|
| Actor Learning Rate | 0.00069 | $3e^{-4}$ |
| Critic Learning Rate | 0.00054 | $3e^{-4}$ |
| Rollout Length | 16 | 32 |
| Number of Epochs | 4 | 64 |
| Number of Minibatches | 16 | - |
| Buffer Size | - | 65536 |
| Batch Size | - | 32 |
| Sample Sequence Length | - | 32 |
| Discount Factor | 0.99 | 0.99 |
| GAE Lambda | 0.95 | 0.95 |
| Clip Epsilon | 0.1 | - |
| Entropy Coefficient | 0.005 | 0.005 |
| Value Function Coefficient | 0.5 | - |
| Max Grad Norm | 0.5 | 0.5 |
| Decay Learning Rates | True | - |
| Standardize Advantages | True | - |
| Number of Simulations | - | 64 |
| Dirichlet Alpha | - | 0.03 |
| Dirichlet Exploration Fraction | - | 0.25 |
| Number of Samples | - | 20 |
| Gaussian Noise Exploration Fraction | - | 0.001 |

---

[8]Available at: https://github.com/EdanToledo/Stoix

Table 5: Hyperparameters for MPO and VMPO

| Parameter | MPO | VMPO |
|---|---|---|
| Rollout Length | 8 | 32 |
| Number of Epochs | 72 | 16 |
| Buffer Size | 200000 | - |
| Batch Size | 256 | - |
| Sample Sequence Length | 16 | - |
| Period | 1 | - |
| Actor Learning Rate | $1e^{-4}$ | $3e^{-4}$ |
| Critic Learning Rate | - | $3e^{-4}$ |
| Dual Learning Rate | $1e^{-3}$ | $1e^{-2}$ |
| $\tau$ (Target Smoothing) | 0.005 | 0.005 |
| Discount Factor | 0.99 | 0.99 |
| Max Grad Norm | 0.5 | 0.5 |
| Decay Learning Rates | True | - |
| Number of Samples | 128 | - |
| $\epsilon_\eta$ | 0.05 | 0.05 |
| $\epsilon_{\alpha_\mu}$ | 0.05 | 0.05 |
| $\epsilon_{\alpha_\Sigma}$ | 0.0005 | 0.0005 |
| Initial $\eta$ | 10.0 | 10.0 |
| Initial $\alpha_\mu$ | 10.0 | 10.0 |
| Initial $\alpha_\Sigma$ | 500 | 500 |
| GAE Lambda | 0.95 | 0.95 |
| Actor Target Period | - | 25 |

Table 6: Hyperparameters for SMC-Ent

| Parameter | SMC-Ent |
|---|---|
| Rollout Length | 16 |
| Number of Epochs | 512 |
| Buffer Size | 500000 |
| Batch Size | 512 |
| Actor Learning Rate | $3e^{-4}$ |
| Q Learning Rate | $3e^{-4}$ |
| $\tau$ (Target Smoothing) | 0.005 |
| Discount Factor | 0.99 |
| Reward Scale | 10.0 |
| Number of Particles | 16 |
| Max Depth | 4 |
| Resample Temperature | 1.0 |
| Number of Samples for Value Function | 64 |

### E.1.2 Discrete Control

Table 7: Hyperparameters for AlphaZero and PPO

| Parameter | AlphaZero | PPO |
|---|---|---|
| Learning Rate | $3e^{-4}$ | $3e^{-4}$ |
| Rollout Length | 21 | 85 |
| Number of Epochs | 16 | 4 |
| Buffer Size | 65536 | - |
| Batch Size | 64 | - |
| Sample Sequence Length | 17 | - |
| Discount Factor | 0.99 | 0.99 |
| GAE Lambda | 0.95 | 0.95 |
| Max Grad Norm | 0.5 | 0.5 |
| Number of Simulations | 64 | - |
| Number of Minibatches | - | 64 |
| Clip Epsilon | - | 0.1 |
| Entropy Coefficient | - | $1e^{-2}$ |
| Standardize Advantages | - | True |

Table 8: Hyperparameters for MPO and VMPO

| Parameter | MPO | VMPO |
|---|---|---|
| Actor Learning Rate | $2e^{-4}$ | $3e^{-4}$ |
| Decay Learning Rates | True | - |
| Dual Learning Rate | 0.02 | 0.01 |
| Number of Epochs | 72 | 4 |
| $\epsilon_\eta$ | 0.07 | 0.5 |
| $\epsilon_\alpha$ | 0.00015 | 0.001 |
| GAE Lambda | 0.95 | 0.95 |
| Discount Factor | 0.95 | 0.99 |
| Initial $\eta$ | 3.0 | 3.0 |
| Initial $\alpha$ | 3.0 | 3.0 |
| Max Grad Norm | 0.5 | 0.5 |
| Number of Samples | 128 | - |
| Q Learning Rate | 0.001 | - |
| Rollout Length | 16 | 86 |
| Sample Sequence Length | 17 | - |
| $\tau$ (Target Smoothing) | 0.005 | - |
| Batch Size | 256 | - |
| Buffer Size | 500000 | - |
| Actor Target Period | - | 64 |

Table 9: Hyperparameters for SMC-Ent

| Parameter | SMC-Ent |
|---|---|
| Rollout Length | 32 |
| Number of Epochs | 64 |
| Buffer Size | 1000000 |
| Batch Size | 2048 |
| Q Learning Rate | $3e^{-4}$ |
| $\tau$ (Target Smoothing) | 0.005 |
| Discount Factor | 0.97 |
| Max Grad Norm | 0.5 |
| Huber Loss Parameter | 1.0 |
| Entropy Temperature | 0.03 |
| Munchausen Coefficient | 0.9 |
| Clip Value Min | $-1e^3$ |
| Number of Particles | 16 |
| Search Horizon | 4 |
| Resampling Period | 4 |
| Resample Temperature | 0.1 |
| Reward Scaling | 10.0 (Rubik's Cube) / 1.0 (Sokoban) |

# F Expectation Maximisation

## F.1 Overview of methods

EM algorithms differ on a small number of dimensions through which the algorithm can be understood. The first dimension, $\mathcal{G}$, is the optimization objective that is maximized in the E-step. This includes whether advantages, Q-values, or rewards are used, and with respect to which policy. We then consider whether a trust region constraint is used both in the E-step and M-step.

With a defined optimization objective, various methods can be used to estimate it. Most methods derive an analytic solution to the optimization problem and then estimate this distribution using techniques such as TD(0) or function approximation. For example, AlphaZero does not explicitly derive the analytic solution but estimates the solution to this optimization objective through Monte Carlo Tree Search (MCTS).

Table 10 provides a summary of some common EM methods, highlighting their core differences.

Table 10: Summary of EM based Algorithms

| Method | $\mathcal{G}$ | E-step TR | M-step TR | $\mathcal{G}$ estimate | Depth | Breadth |
|--------|------|-----------|-----------|------------|-------|---------|
| MPO | $Q^{\pi_p}$ | Yes | Yes | Analytic + Function Approximation | 0 | M |
| V-MPO | $A^{\pi_p}$ | Yes | Yes | Analytic + n-step TD + top-k Adv | $T$ | 1 |
| AWR | $A^{\pi_p}$ | No | No | Analytic + n step TD | T | 1 |
| AWAC | $A^{\pi_p}$ | No | No | Analytic + Function Approximation | 0 | 1 |
| PoWER | $\eta \log Q^{\pi_p}$ | Yes | No | Analytic + n step TD | T | 1 |
| RWR | $\eta \log r$ | No | No | - | 1 | 1 |
| REPS | $A^{\pi_p}$ | Yes | No | Analytic + TD(0) | 1 | 1 |
| AlphaZero | $Q^{\pi_q}$ | Yes | No | MCTS | $>0$ | $>0$ |
| SPO | $A^{\pi_q}$ | Yes | Yes | Analytic + SMC | $D$ | M |

Note that $T$ refers to an episode length. In additon we add two important dimensions of the $\mathcal{G}$ estimate (breadth and depth). Methods like MPO estimate the analytic solution directly using a Q-function. This can be leveraged to estimate the target distribution for a selection of $M$ actions, but without leveraging depth, so rewards and future states are not used to improve the estimates. In contrast, V-MPO leverages depth to form an n-step TD estimate, but only for one of the actions, leading to a relatively poor estimate of the analytic solution.

# G    Proofs and Discussions

## G.1    E-step Analytic solution

We outline the analytic solution to eq. (5), by writing a Lagrangian equation and solving for $q$. We can first represent our constrained optimisation exactly with the following optimisation problem and associated 2 constraints. We add a state dependent baseline $V(s)$ to the optimisation objective and notate $Q^q(s,a) - V^q(s)$ as $A^q(s,a)$.

$$
\max_q \int \mu_q(s) \left[ \int q(a|s) \left[ Q^q(s,a) - V^q(s) \right] da \right] ds
$$
$$
\text{s.t.} \int \mu_q(s) \left[ \text{KL}(q(a|s) \| \pi(a|s, \theta_i)) \right] ds < \epsilon, \tag{13}
$$
$$
\int \mu_q(s) \left[ \int q(a|s) da \right] ds = 1.
$$

The following Lagrangian $L$ can be constructed,

$$
L(q, \eta, \gamma) = \int \mu_q(s) \left[ \int q(a|s) A^q(s,a) \, da \right] ds
$$
$$
+ \eta \left( \epsilon - \int \mu_q(s) \left[ \int q(a|s) \log \left( \frac{q(a|s)}{\pi(a|s, \theta_i)} \right) da \right] ds \right) \tag{14}
$$
$$
+ \gamma \left( 1 - \int \mu_q(s) \left[ \int q(a|s) \, da \right] ds \right).
$$

We can then solve for the value of $q$ that maximises this expression by taking the derivative with respect to $q$ and setting it to 0.

$$
\frac{\partial L(q, \eta, \gamma)}{\partial q} = A^q(s,a) - \eta \log q(a|s) + \eta \log \pi(a|s, \theta_i) - (\eta + \gamma), \tag{15}
$$

The analytic form for the optimal distribution $q$ can then be calculated as:

$$
q(a|s) = \pi(a|s, \theta_i) \exp \left( \frac{A^q(s,a)}{\eta} \right) \exp \left( -\frac{\eta + \gamma}{\eta} \right). \tag{16}
$$

## G.2    E-step KL constraining temperature

We now derive the dual function to be minimised in order to obtain $\eta$ within the analytic solution for $q$, which is crucial for enforcing the KL constraint.

The final term $-\frac{\eta + \gamma}{\eta}$ acts as a normalising constant since it is independent of $q$ and therefore we can construct the following equality.

$$
\exp \left( \frac{\eta + \gamma}{\eta} \right) = \int \pi(a|s, \theta_i) \exp \left( \frac{A^q(a,s)}{\eta} \right) da, \tag{17}
$$

Now that we have expressions for $-\frac{\eta + \gamma}{\eta}$ and $q$ we can form the dual function $g(\eta)$ by substituting these terms back into the original Lagrangian. After simplifying we recover the following expression.

$$
g(\eta) = \eta \epsilon + \eta \int \mu_q(s) \log \left( \int \pi(a|s, \theta_i) \exp \left( \frac{A^q(a,s)}{\eta} \right) da \right) ds \tag{18}
$$

The optimal dual variable can be calculated as follows

$$
\eta^* = \arg \min_\eta g(\eta). \tag{19}
$$

### G.3 E step: Constraint Optimisation

In optimising eq. (4), the first term $\mathbb{E}_{q(a|s)}[Q^q(s,a)]$ is dependent on the scale of reward in the environment. This can make it hard to choose $\alpha$, as the first and second terms are on arbitrary scales. Practically this can mean that for each new environment a new $\alpha$ should be used requiring costly hyperparameter tuning to find. As explored in previous work [1, 76] we choose to enforce a hard constraint, as opposed to a soft constraint. Instead of choosing $\alpha$ we choose $\epsilon$ which is the maximum KL divergence between $q$ and $\pi$ which is practically less sensitive to reward scale. We found that choosing an $\epsilon$ to be far easier, resulting in a value that generalised across all environments explored. This is important as there has been a trend in recent years to add hyperparameters to Reinforcement Learning algorithms [2] resulting in the need for costly hyperparameter sweeps for algorithms to work on each new problem they are applied to.

### G.4 Proof of Proposition 1: Monotone Improvement Guarantee

The optimality of policy $\pi_\theta$, $\log p_{\pi_\theta}(\mathcal{O} = 1)$, is lower bounded by the following ELBO objective:

$$\mathcal{J}(q, \pi_\theta) = \mathbb{E}_{\tau \sim q}\left[\sum_{t=0}^{\infty}\left(\gamma^t r_t - \alpha D_{KL}(q(\cdot|s_t)\|\pi(\cdot|s_t, \theta))\right)\right] + \log p(\theta).$$

We improve $\pi_\theta$ by optimizing the ELBO alternatively via EM. We will outline the proof of the monotonic improvement guarantee of the ELBO using the EM procedure, at the $i$-th iteration of training. We note that this holds under the assumption that we calculate the true value of the closed form solution $q_i$ in the E-step, which is in practice unlikely to be the case.

**E-step:** By the definition of E-step, we improve the ELBO with respect to $q$. We show the E-step update will increase ELBO:

$$q_{i+1} = \arg\max_q \mathbb{E}_q\left[\mathbb{E}_{a \sim q(\cdot|s)}[A^{q_i}(s,a)] - \alpha D_{KL}[q(\cdot|s)\|\pi(\cdot|s,\theta_i)]\right]$$

$$= \arg\max_q \mathbb{E}_{\tau \sim q}\left[\sum_{t=0}^{\infty}\left(\gamma^t r_t - \alpha D_{KL}(q(\cdot|s_t)\|\pi(\cdot|s_t, \theta_i))\right)\right]$$

$$= \arg\max_q \mathcal{J}(q, \theta_i)$$

$$\Rightarrow \mathcal{J}(q_{i+1}, \pi_{\theta_i}) \geq \mathcal{J}(q_i, \pi_{\theta_i}).$$

**M-step:** We update $\theta$ by

$$\theta_{i+1} = \arg\max_\theta \mathbb{E}_{q_{i+1}}\left[\alpha \mathbb{E}_{a \sim q_{i+1}(\cdot|s)}[\log \pi_\theta(a|s)] + \log p(\theta)\right]$$

$$= \arg\max_\theta \mathbb{E}_{q_{i+1}}\left[-\alpha D_{KL}[q_{i+1}(\cdot|s_t)\|\pi(\cdot|s_t, \theta)] + \log p(\theta)\right]$$

$$= \arg\max_\theta \mathbb{E}_{q_{i+1}}\left[\mathbb{E}_{a \sim q_{i+1}(\cdot|s)}[A^{q_i}(s,a)] - \alpha D_{KL}[q_{i+1}(\cdot|s_t)\|\pi(\cdot|s_t, \theta)] + \log p(\theta)\right]$$

$$= \arg\max_\theta \mathcal{J}(q_{i+1}, \pi_\theta).$$

Therefore, we have: $\mathcal{J}(q_{i+1}, \pi_{\theta_{i+1}}) \geq \mathcal{J}(q_{i+1}, \pi_{\theta_i})$. Combining these two results we have that after successive applications of the E-step and M-step,

$$\mathcal{J}(q_{i+1}, \pi_{\theta_{i+1}}) \geq \mathcal{J}(q_i, \pi_{\theta_i}).$$

### G.5 M-step objective

In Reinforcement Learning it can be beneficial to constrain policies from moving too far from the current policy, often leading to increased stability or performance of an algorithm [69, 72]. In this work we utilise a Gaussian prior around $\theta_i$, our current value of $\theta$, optimised from the previous iteration.

Therefore, $\theta \sim \mathcal{N}(\mu, \Sigma)$ where $\mu = \theta_i$ and $\Sigma^{-1} = \lambda F(\theta_i)$, where $F$ is the Fisher information matrix. The prior term in eq. (11) can then be written as $\log p(\theta) = -\lambda(\theta - \theta_i)^T F(\theta_i)^{-1}(\theta - \theta_i) + c$. The first term in this expression is the quadratic approximation of the KL divergence [69] while the second term $c$ is a term that does not depend on $\theta$ and therefore when optimising eq. (11) with respect to $\theta$, can be dropped. Using this approximation we rewrite eq. (11) as:

$$\max_\theta \mathbb{E}_{s \sim \mu_q(s)}\left[\mathbb{E}_{a \sim q(a|s)}[\log \pi(a|s, \theta)] - \lambda \text{KL}(\pi(a|s, \theta_i) \| \pi(a|s, \theta))\right]$$

Like the E-step, choosing $\lambda$ can be non-trivial so we convert it into a hard constraint optimisation eq. (12). Note that $\epsilon$ in eq. (12) is different from eq. (5). We note that considering a Gaussian prior in the M-step is not strictly required for SPO performance, however practically it adds stability to training.

## G.6 Connection to Mirror Descent Guided Policy Search

We highlight the connection between the use of Expectation Maximisation to improve the evidence lower bound and Mirror Descent Guided Policy Search (MD-GPS). Mirror Descent Guided Policy Search builds upon previous guided policy search work [46]. Rather than only enforcing the constraint between $q$ and $\pi$ at convergence (referred to a the local policy:$p_i$ and global policy: $\pi_\theta$ respectively), they constrain $q_i$ against the current policy $\pi_i$ at every iteration. This results in a very similar algorithm to EM optimisation. Below we provide the outline of MD-GPS algorithm, clearly showing the equivalence to EM [58]:

---

**Algorithm 3** Mirror descent guided policy search (MDGPS): convex linear variant

---

1: **for** iteration $k \in \{1, \dots, K\}$ **do**
2:     **C-step:** $p_i \leftarrow \arg\min_{p_i} \mathbb{E}_{p_i(\tau)} \left[ \sum_{t=1}^{T} \ell(\mathbf{x}_t, \mathbf{u}_t) \right]$ such that $D_{\text{KL}}(p_i(\tau) \parallel \pi_\theta(\tau)) \leq \epsilon$
3:     **S-step:** $\pi_\theta \leftarrow \arg\min_\theta \sum_i D_{\text{KL}}(p_i(\tau) \parallel \pi_\theta(\tau))$
4: **end for**

---

Comparing MD-GPS to EM, the C-step is equivalent to the E-step and S-step equivalent to the M-step, with the loss $\ell$ being the negative of the expected discounted sum of returns over trajectories. MD-GPS assumes that the S-step can perfectly minimise the objective in the case of linear dynamics and quadratic costs [48], resulting in exact mirror descent. In practice, most applications of MD-GPS will not satisfy such constraints and so are akin to approximate mirror descent. However, as long as a constraint between the local and global policy can be enforced in terms of KL, various bounds on cost of the global policy can be constructed, see Montgomery and Levine [58] for further details.

# H Statistical Precipice

## H.1 Overview

Advancements in computational power and algorithmic capabilities have led to a shift in the evaluation of reinforcement learning algorithms, now typically assessed through extensive suites of tasks. Performance metrics such as the mean or median score per task are commonly used, but these metrics can fail to account for the statistical uncertainties arising from a limited number of runs and varying random seeds. The trend towards computationally intensive benchmarks further complicates the issue, as each run can span from hours to weeks, making it impractical to conduct numerous runs per task and thereby increasing the uncertainty in the reported metrics. To address these challenges, we adopt the evaluation methodologies proposed by Agarwal et al. [3] and Gorsane et al. [32].

Each algorithm is evaluated across $M$ tasks within a specified environment suite, with $N$ independent runs per task $m \in M$. During each run $n \in N$, performance is measured over $E$ episodes, each consisting of $T$ timesteps. At each interval $i$, the mean return $G^i_{m,n}$ is computed, and the model is checkpointed. The model with the highest mean return across all intervals is selected for final evaluation.

Following the completion of a training run $n \in N$, the best model is further evaluated over $10 \times E$ episodes. We normalise scores $x_{m,n}$ for each task $m = 1, \ldots, M$ and run $n = 1, \ldots, N$, scaling them based on the minimum and maximum scores observed across all runs. This normalization produces a set of normalised scores $x_{1:M,1:N}$ per algorithm. These scores are then aggregated into a single scalar estimate, $\bar{x}$.

To ensure robust statistical confidence, we employ a 95% confidence interval derived from stratified bootstrapping over the $M \times N$ experiments, treating these as random samples. This method integrates the performance across all tasks and runs, simulating the statistical reliability of multiple runs on a single task while considering task diversity. Results are reported for the entire suite.

**Metrics**:

We utilize normalised scores to evaluate algorithm performance, employing metrics that go beyond simple median and mean calculations:

- **Interquartile Mean (IQM):** This metric calculates the mean of the central 50% of runs, excluding the lower and upper 25%. It is more robust to outliers than the mean and less biased than the median, offering higher statistical efficiency and detecting improvements with fewer runs [3].
- **Probability of Improvement:** This measures the likelihood that algorithm X will outperform algorithm Y on a random task $m$, using the Mann-Whitney U-statistic. It is defined as:

$$Pr(X > Y) = \frac{1}{M} \sum_{m=1}^{M} Pr(X_m > Y_m) \tag{20}$$

$$Pr(X_m > Y_m) = \frac{1}{NK} \sum_{i=1}^{N} \sum_{j=1}^{K} S(x_{m,i}, y_{m,j}) \tag{21}$$

$$S(x, y) = \begin{cases} 1, & \text{if } y < x \\ \frac{1}{2}, & \text{if } y = x \\ 0, & \text{if } y > x \end{cases} \tag{22}$$

Statistical significance is determined using the Neyman-Pearson criterion, based on the confidence interval bounds [13].

Additionally, performance profiles can be employed to visually compare methods, illustrating the fraction of runs that exceed a given threshold. These profiles aid in identifying stochastic dominance and empirical performance bounds. We also plot the interquartile mean score against environment steps to evaluate sample efficiency.

## H.2 Hyperparameters

Table 11: Statistical Precipice Evaluation Hyperparameters

| Parameter | Value |
| --- | --- |
| Number of Games for In-Training Evaluation - $E$ | 128 |
| Number of Games for End-of-Training Evaluation - $10 \times E$ | 1280 |
| Seeds per environment - $N$ | 5 |
| Tasks per suite - $M$ | Continuous=3, Discrete=2 |

