# OpenReview forum: "SPO: Sequential Monte Carlo Policy Optimisation"
_NeurIPS.cc/2024/Conference — NeurIPS 2024 poster_

### Official Review · Reviewer_X2nr · 2024-06-28

**Soundness:** 2
**Presentation:** 3
**Contribution:** 3
**Rating:** 5
**Confidence:** 3

**Summary:**

This paper maximizes the expected reward wrt the policy by rewriting it as a log marginal likelihood in a probabilistic model where latents comprise of states, actions, and “optimality” observations.

The prior over states is the world model and the prior over actions is the policy which being optimized. The likelihood is the exponentiated reward which can be normalized to make it a distribution over auxiliary “optimality” variables.

The marginal likelihood can be optimized via EM where in the E step, we maximize the evidence lower bound (ELBO) in (3) with respect to the variational distribution q and in the M step, we fix q and maximize the objective with respect to the policy π.

In the E-step (Sec. 4.2), the paper derives an analytic expression for q via the Lagrange multipliers method (eq. 6).

In the M-step (Sec. 4.3), the paper uses this q to maximize the objective with respect to π. The key difficulty is that the q derived in the E-step is only known up to a normalizing constant and so the paper proposes to get approximate samples from q using SMC (Sec. 4.2.1).

The experiments compare the proposed method to baselines on several tasks with discrete and continuous action spaces, and show wins. There are ablations showing the tradeoff between using more particles vs unrolling for longer.

In addition to performance, the paper argues that SPO is preferable to the popular MCTS method because it parallelizes more easily and performs well across more domains.

**Strengths:**

Combining SMC and RL is interesting and the experiments show strong empirical gains. However, theoretically, I am not fully satisfied with the method itself—or at the very least, I found it difficult to understand the method fully from the description. See questions / comments below.

Therefore, I’m giving this paper a reject in its current form but I am willing to raise the score if the method or its description is improved.

**Weaknesses:**

-

**Questions:**

Major
- SMC (Sec. 4.2.1): From (6) it seems like there is just one distribution we want to sample from instead of a sequence of them. What’s the sequence of distributions that SMC targets? In other words, in the background section about SMC, it’s unclear what is $p(x)$ and what is $p(x_t | x_{1:t - 1})$; how do terms in (8) relate to this $p$?
- If I reverse-engineer a sequence of targets distribution from (8), it should be $p(a_{1:t}, s_{1:t}) = \prod_{j = 1}^t \mathcal T(s_t | …) \pi(a_t | …) \exp(A^{\bar \pi}(a_t, s_t) / \eta^*)$. That is, the final target distribution is the transition times the policy times the exponent of the *sum* of advantage functions. I think this is incorrect since it doesn’t correspond to the sum of per-step rewards. Something that would make more sense to me is to have the incremental weights contain the exponentiated rewards for all time steps instead of the last one, where it’s the exponentiated advantage function. This is obviously a completely different algorithm.
- I don’t understand why (5) follows from (4). We have an unconstrained optimization of a regularized objective in (4) which is turned into a constrained optimization problem in (5). I also don’t understand how $\alpha$ turns into $\epsilon$.
- Why does (10) rewrite (9) as a constrained objective where it seems like in practice, we’re actually just optimizing (9).
- How is the value function and the Q-function learned? These seem critical for the method but I couldn’t find a description.
- E-step: I agree that it is difficult to optimize with respect to $q$ since it’s the distribution of the expectation and the term inside the expectation, $Q^q$ depends on it (eq. 4). The paper chooses to fix $Q^q$. I would have expected that it is the $q$ in $\mathbb E_q$ that makes optimization difficult since typically we’d have to resort to the reparameterization trick or REINFORCE. If we fix the $q$ in $\mathbb E_q$, optimizing the expectation is simple: just sample from $q$ and directly optimize $Q^q$.

Minor
- $\gamma$ in (3) is not introduced before.
- Should there be a likelihood term in the incremental weight in (2)? Or is $p(x_t | x_{1:t - 1})$ an unnormalized distribution? $p$ is not defined in this paragraph.
- Swapping the prior term $\log p(\theta)$ in (9) to the KL term in (10) is a little odd. Would it simply not work otherwise?
- In App. G.1, why is $\mu_q$ dependent on $q$?

Other relevant work
- [AESMC](https://arxiv.org/abs/1705.10306), [FIVO](https://arxiv.org/abs/1705.09279), [VSMC](https://arxiv.org/abs/1705.11140)
- [DVRL](https://arxiv.org/abs/1806.02426)
- [SIXO](https://arxiv.org/abs/2206.05952), [NAS-X](https://arxiv.org/abs/2308.14864)

**Limitations:**

-

---

> ### Author Rebuttal · Authors · 2024-08-01
>
> We thank the reviewer for their detailed review which will be used to clarify details of the paper
>
> 1. We aim to sample from the target policy over sequences $\tau= (s_0, a_0, s_1, a_1, ..., s_t, a_t, s_{t+1})$  similar to [60]. In the following notation we switch to using $\tau$ from $x$ and give definitions for the density $p$ and proposal $\beta$ at policy iteration $i$.
>
> * $p_i(\tau_{1:t})  \propto \mu(s_0) \prod_{j=0}^t \mathcal{T}(s_{j+1}|s_j,a_j) \pi_i(a_j|s_j, \theta_i) \exp\left(\frac{\bar{A_i}(a_j,s_j)}{\eta^*_i}\right)$
>
>
> * $p_i(\tau_t|\tau_{1:t-1}) \propto \mathcal{T}(s_{t+1}|s_{t},a_{t}) \pi_i(a_t|s_t, \theta_i) \exp\left(\frac{\bar{A_i}(a_t, s_t)}{\eta_{i}^*}\right)$
>
> * $\beta(\tau_{1:t})  \propto \mu(s_0) \prod_{j=0}^t \mathcal{T_{\text{model}}}(s_{j+1}|s_j,a_j) \pi_i(a_j|s_j, \theta_i)$
>
> * $\beta(\tau_{t}|\tau_{1:t-1})  \propto \mathcal{T_{\text{model}}}(s_{j+1}|s_j,a_j) \pi_i(a_j|s_j, \theta_i)$
>
> The weight update:
>
> $$w_t(\tau_{1:t}) \propto w_{t-1}(\tau_{1:t-1}) \cdot \frac{p(\tau_t|\tau_{1:t-1})}{\beta(\tau_t|\tau_{1:t-1})} = w_{t-1}(\tau_{1:t-1}) \cdot \left(\frac{\mathcal{T}(s_{t+1}|s_{t}, a_{t})}{\mathcal{T_{model}}(s_{t+1}|s_{t}, a_{t})}\right) \cdot \frac{\exp(\bar{A_i}(a_t, s_t)/\eta_{i}^*) \cdot \pi_i(a_t|s_t, \theta_i)}{\pi_i(a_t|s_t, \theta_i)}$$
>
> 2. You are correct, the target distribution over sequences utilises the sum of advantages and not explicitly per step rewards. This does account for the sum of per step rewards, as we calculate $A^{\pi}(a_t,s_t) = E_{\pi}[r(a_t,s_t) + \gamma V(s_{t+1})] - V(s_t)$ therefore for $\gamma = 1$, over sequences, we have the following estimate $\sum_{t=0}^h A^{\pi}(a_t,s_t) = \sum_{t=0}^h r(a_t,s_t) + V^{\pi}(s_{h+1}) - V^\pi(s_{0})$,
> the sum of per step rewards plus the value of the last state, minus the value of the first state. If the weight only uses the per step rewards, future rewards beyond the planning horizon are not accounted for. This destroys performance, see new results fig 2. Without a baseline, the weight update would be $\prod_{j=0}^t \exp(Q^{\pi}(a_t,s_t)/\eta^*)$, see line 660 for ablation.
>
> 3. In optimising 4, the first term $\mathbb{E}_{q(a|s)} \left[Q^q(s,a)\right]$ is dependent on the scale of reward in the environment. This can make it hard to choose $\alpha$, as the first and second terms are on arbitrary scales. As explored in previous work [1,69] we choose to change this from a soft constraint to a hard constraint (line 158). Instead of choosing $\alpha$ we choose $\epsilon$ which is the maximum KL divergence between $q$ and $\pi$ which is practically less sensitive to reward scale.
>
> 4. Eq 9 is a supervised learning step, we have choice over the parameterisation of our policy and prior for regularisation. RL algorithms often constrain policies from moving too far from the current policy, in this work we utilise a Gaussian prior around $\theta_i$ optimised from the previous iteration. Therefore, $\theta \sim \mathcal{N}(\mu, \Sigma)$ where  $\mu = \theta_i$ and $\Sigma^{-1} = \lambda F(\theta_i)$, where $F$ is the Fisher information matrix. The prior term in (9) can written as $\log p(\theta) = -\lambda(\theta - \theta_i)^T F(\theta_i)^{-1}(\theta - \theta_i) + c$.  Note $c$ is a term that do not depend on $\theta$. The first term is the quadratic approximation of the KL divergence [2*], so we can rewrite equation (9) as:
>
> $$
> \max_{\pi} \mathbb{E}\_{\mu_q(s)} \left[ \mathbb{E}_{q(a|s)} \left[ \log \pi(a|s, \boldsymbol{\theta}) \right] - \lambda \text{KL} \left( \pi(a|s, \boldsymbol{\theta_i}), \pi(a|s, \boldsymbol{\theta}) \right) \right]
> $$
>
> Like the E-step, choosing $\lambda$ can be non-trivial so we convert it into a hard constraint optimisation (eq 10). $\epsilon$ in Eq (10) is different from (5). The constrained M-step (with a Gaussian prior) is not required for SPO, however it adds stability to training.
>
> 5. SPO does not require a Q-function and only learns a value function since we leverage observed rewards and state-values from planning to calculate advantages. We learn the value function like PPO, i.e using GAE values. GAE combines multiple N-step td-errors estimates,
> $A_\text{GAE}(s_t) = \sum_{l=0}^{N} (\gamma \lambda)^l \delta_{t+l}$
> where $\delta_t = r_t + \gamma V(s_{t+1}) - V(s_t)$. Such that $V_{target}(s_t) = V(s_t, \theta) + A_\text{GAE}(s_t)$. These value targets are not constructed during planning but on real rollouts. It is a practical choice as to what method is used to generate targets, we opt for GAE to balance bias and variance.
>
> 6. In Eq 4 when optimising with respect to q there is a moving target, each update to q results in a new critic $Q^q$. Our approach aligns with most actor-critic approaches [65] that fix a critic within an iteration of policy optimisation. For the suggestion of optimising the $Q^q$ term and fixing the acting policy  $q$, this would be a highly non-trivial optimisation. Since $Q^q$ depends on the transition dynamics and reward function of the environment unless we knew these we would need to perform off-policy evaluation over the space of possible policies $q$, with such evaluation known to be extremely challenging [1*], as behaviour of policies is difficult to estimate using different distributions. We are not aware of RL algorithms that approach optimisation from this perspective.
>
> **Minor**
> 1. $\gamma$ is introduced in the background as the discount factor however we will add to 3) for clarity
>
> 2. $p(x_t| x_{1:t-1})$ is unnormalised, we define p(x) before eq 2) as an arbitrary target distribution
>
> 3. See major q4
>
> 4. $\mu$ is the distribution over states. Any policy acting in the MDP will induce a different state distribution. We therefore denote $\mu_q$ in Eq 11 as this relies on the policy $q$. We will add this clarification to the manuscipt.
>
> **References**
>
> [1*]  Dudık et al "Doubly Robust Policy Evaluation and Learning" ICML 2011
>
> [2*] Schulman et al "Trust region policy optimization" International conference on machine learning PMLR 2015

---

### Official Review · Reviewer_xUvG · 2024-07-11

**Soundness:** 2
**Presentation:** 1
**Contribution:** 2
**Rating:** 4
**Confidence:** 3

**Summary:**

This paper propose SPO:  Sequential Monte Carlo Policy Optimisation, a RL algorithm with the Expectation Maximisation (EM) framework for MDP. Experiments on both discrete and continuous environments show that SPO outperforms baselines in terms of sample efficiency and scalability.

**Strengths:**

Quality: The paper clearly introduces the background of Sequential decision-making, Control as inference, Expectation Maximisation and Sequential Monte Carlo.
Significance: The performance improvement of the proposed algorithm is significant.

**Weaknesses:**

Originality: The paper propose SPO, which is the combination of Sequential Monte Carlo (SMC) sampling with the EM framework. The novelty is limited.
Quality: The paper does not discuss what problem the paper aims to tackle. The paper uses SMC to better estimate the target distribution. But the paper does not give any experimental results showing better estimation results. Ablation studies are missing. Theoretical insights of proposition 1 are missing.
Clarity: The paper is hard to follow as it is full of formulas.

**Questions:**

See the weakness.

**Limitations:**

Yes.

---

> ### Author Rebuttal · Authors · 2024-08-05
>
> **On Novelty:** Re: "limited novelty'', we respectfully disagree with the reviewer. We highlight three contributions (all included in the manuscript lines 27, 39, 38):
>
> 1. This is the first instance of Sequential Monte Carlo being used as a policy improvement operator for RL. To date, MCTS has been the most widely adopted search-based policy improvement operator. The paper presents a robust search algorithm, with a theoretical foundation, resulting in an operator that outperforms competing methods on a range of benchmarks.
> 2. The introduction of an inherently parallelisable operator is also a strong contribution. We are not aware of any successful search based policy improvement operations that are able to leverage parallelism over search to speed up inference during training. In fact most works note that parallelisable versions of MCTS strongly reduce performance [1,2] and have not successfully implemented parallelisable versions of AlphaZero for policy improvement[3].
> 3. Introducing a method that requires no algorithmic modifications between continuous and discrete environments while demonstrating strong performance in both settings, should not be considered as limited.  The generality of popular RL algorithms is often limited with algorithms such as AlphaZero requiring considerable modifications just to be applied to complex continuous domains. This is an important barrier for the applicability of search based policy
>
> **On Quality/Clarity:**
> The manuscript highlights deficiencies with the current SOTA methods (lines 22-25). After introducing SPO (lines 26-36), we explain how SPO tackles each of the highlighted problems. To clarify, we aim to solve the scalability problem in current search based policy improvement methods, such that search can be executed in parallel without performance deterioration observed in previous works [1,2]. This allows for search methods to leverage hardware accelerators to perform search faster reducing train times (lines 38-39). We explain that SPO is widely applicable to discrete and continuous environments without modifications (lines 37-39). Lastly we explain SPO is theoretically backed  (line 29) and that the method scales for different levels of planning (line 41-42).
>
>
>
> **On Ablations:** "Ablation studies are missing'' This is an incorrect statement from the reviewer. In the original manuscript we provide two important ablations. We ablate the difference between sampling the target policy in (6) and the alternative formulation without a baseline (line 660), important as previous works differ on these objectives [1,69]. We also provide a second ablation that demonstrates how the appropriate temperature for SMC varies significantly between environments (line 649). This backs up our approach of learning the temperature variable, and demonstrates this is an important contribution to reduce hyperparamter tuning. We add a 3rd ablation to elicit insight into how SPO enforces the KL constraint in policy improvement (Figs 1). We show two charts with varying runs for different maximum KL constraints. We see when viewed on the iteration level that SPO is able to accurately target the KL of choice (fig 1 a). Fig 1 b) shows the impact different KL constraints have on training curves.
>
> Re: "the paper does not give experimental results showing better estimation results". In practice we do not have access to the target distribution, therefore it is non-trivial to construct an ablation that directly measures how good methods are at estimating it. However, we add new ablation that leverages Monte Carlo samples to generate an unbiased estimate (given large compute budget). It compares the KL divergence of our SMC policy to a Monte Carlo oracle (Fig 3). We investigate the impact of depth and particle number on the KL divergence to the oracle for SMC on Sokoban. This concludes that scaling particles is particularly important for improving the target estimation, aligning with SMC theory [4], also that leveraging depth improves estimation.  Methods like MPO that limit particles to the number of actions and depth 0 are severely limited in their estimation of the target, likewise V-MPO only uses 1 sample albeit to a depth of N. Lastly comparing MPO to SPO there is a performance gap across all environments. Both methods have similar theoretical foundations the difference being how samples are obtained from the target. Therefore any performance differences can be attributed to changes in target estimation.
>
> **On Theoretical insights:** Similarly, the following statement made by the reviewer is untrue
> "Theoretical insights of proposition 1 are missing" We provide full theoretical insights for proposition 1 in Appendix section G.3 (line 749). This is referenced in the main manuscript on line 238.
>
> **On Formulas:** Many of the formulas are either existing work, or modifications to existing work. We believe that the provided theoretical justification is in line with the standards of rigour expected at NeurIPS. It is reasonable to expect that experts in the field would feel comfortable with work that provides such level of detail in order to clearly understand the theoretical foundations and methodology. If you think certain formulas could be re-positioned in the paper, with additional insights provided, we welcome specific feedback in order to increase readability.
>
> **References**
>
> [1] Liu, Anji, et al. "Watch the unobserved: A simple approach to parallelizing monte carlo tree search" arXiv preprint (2018)
>
> [2] Segal Richard B. "On the scalability of parallel UCT." International Conference on Computers and Games. Berlin, Heidelberg: Springer Berlin Heidelberg 2010
>
> [3] Seify Arta, and Michael Buro. "Single-agent optimization through policy iteration using monte-carlo tree search." arXiv preprint  (2020)
>
> [4] Del Moral et al Branching and interacting particle systems approximations of Feynman-Kac formulae with applications to non-linear filtering Springer Berlin Heidelberg 2000

---

### Official Review · Reviewer_R8DZ · 2024-07-11

**Soundness:** 3
**Presentation:** 3
**Contribution:** 3
**Rating:** 7
**Confidence:** 4

**Summary:**

This paper introduces a novel iterative, particle-based approach to sequential Monte Carlo planning by combining expectation maximization with importance sampling. Their approach uses a model to sample real state-action trajectories in a problem domain then, after importance-weighting said transitions, computes a target policy distribution for the agent. This target policy is used for policy improvement. The paper supplies a theoretical derivation of their approach and provides experimental results against baseline algorithms.

**Strengths:**

**Originality**: The paper introduces a novel combination of previously established algorithms to tackle sequential decision making.

**Significance:** The results of the paper demonstrate that their proposed approach is *easily parallelizable*, *per-iteration faster* and *explicitly tackles a shortcoming of a prior approach (i.e., addresses weight degeneracy)*. All these would make their work interesting to the planning community.

**Clarity:** The paper is well written and easy to understand.

**Quality:** The motivation for the work is strong as they study a problem of general interest to the community. The conceptual rationale for their approach is sound and the theoretical build makes sense. The experimental evaluation focussed on the claims the authors made and back up said claims. They not only show performance improvements against commonly used approaches but also improvements on computational costs.

**Weaknesses:**

I think that an evaluation with an imperfect model should have been included rather than left as future work.

**Questions:**

Line 204: How exactly is $\hat{A}$ computed? It isn't very clear on that.

How much does removing samples based on weight affect exploration? Is this only helpful when you have a perfect model?

**Limitations:**

They broach the limitations of their work sufficiently.

---

> ### Author Rebuttal · Authors · 2024-08-05
>
> We thank the reviewer for their  positive review.
>
> Re: evaluation of SPO with an imperfect model. Our primary goal was to focus on the novel planning aspects of SPO and its use within training. We believe that the results demonstrate that SMC is a valuable planning method in its own right demonstrably outperforming other commonly used planning methods like AlphaZero when the model is known.
>
> However, we acknowledge that incorporating a learned model is a crucial next step for demonstrating practical applicability. Recent advancements in model-based methods, such as DreamerV3 [2] and EfficientZeroV2 [3], offer advanced architectures for modelling the underlying MDP. These methods are policy-learning agnostic and could easily integrate with the SPO algorithm. As long as the world model accurately predicts rewards, SPO should perform effectively. Additionally, Piché et al. (2018) [1] demonstrated the feasibility of using a learned model with a SMC-based search for maximum entropy RL. This prior work supports the potential integration of learned models in our approach, which we plan to explore in future work.
>
> **Questions**
>
> *Q: Line 204: How exactly is $\hat{A}$ computed? It isn't very clear on that.*
>
> A: Practically, at each step in the sequence we estimate the advantage of a single state action pair using a 1-step estimate $\hat{A}(a_t,s_t) = r(a_t,s_t) + \gamma V(s_{t+1}, \theta_i) - V(s_t, \theta_i)$. As we perform search this we accumulate advantages at each step to calculate the overall importance weights of the sequence according to Eq 8. However, if desired, any advantage estimation technique that can be constructed during the search from a sequence of rewards and values. For example, GAE to allow a trade-off between bias and variance.
>
>
> *Q: How much does removing samples based on weight affect exploration? Is this only helpful when you have a perfect model?*
>
> A: This is a good question. The most important factor is actually the temperature variable which is controlled by the desired KL divergence $\epsilon$ between the target policy $q_i$ generated by search and the current policy $\pi_i$. If the temperature is too low, then during resampling, the particles can collapse onto a single root action which would make the rest of the search pointless. But when the temperature is too high, the search can be too exploratory and not effectively travel down promising paths. This does need to be considered when scaling the search method depth-wise and when choosing the resample period. This wouldn't only be helpful when using a perfect model as ultimately as long as a learned model has accurate reward prediction and dynamics prediction, the exploration/exploitation acts the same. We will amend our manuscript to include this explanation in order to provide greater intuition on the exploration of the SPO search.
>
> Thank you once again for your valuable feedback.
>
> **References**
>
> [1] Piché, Alexandre, et al. "Probabilistic planning with sequential monte carlo methods." International Conference on Learning Representations. 2018.
>
> [2] Hafner, Danijar, et al. "Mastering diverse domains through world models." arXiv preprint arXiv:2301.04104 (2023).
>
> [3] Wang, Shengjie, et al. "EfficientZero V2: Mastering Discrete and Continuous Control with Limited Data." arXiv preprint arXiv:2403.00564 (2024).

---

> > ### Comment · Reviewer_R8DZ · 2024-08-12
> > **Acknowledgement of the rebuttal**
> >
> > I acknowledge that I have read the rebuttal and other reviews and I maintain my score.

---

### Author Rebuttal · Authors · 2024-08-06

In this section we provide additional results/experiments in response to reviewer xUvG and reviewer X2nr.

**Additional Results**

**Figure 1**

While we give in depth details regarding how the KL constraint arises within SPO and the method through with it is enforced (see Appendix G.2). In the original manuscript we did not provide concrete evidence of this KL control aside from showing how important the temperature can be for training (see ablation B.1). Therefore we add an ablation that plots both the performance of various SPO training runs with different values of $\epsilon$ ,on Brax, but also an iteration level mean of the KL divergence between the $\pi_i$ and $\hat{q}$, the current policy and estimate of the target $q$. We expect this value to stay very close to the target, as for each iteration a new temperature value is calculated to directly target this measure. The results show two things, firstly that our method of KL control is very successful and stable. Secondly it shows the importance of KL divergence in convergence in performance.

**Figure 2**

In response to queries regarding the weight update we use in SPO, we choose to add an additional experiment to the ablation in section B.2 . We remove the use of the action independent baseline, but also the use of the bootstrapped value function. This results in the weight update simply according to observed rewards. The provided results in figure 2 show that not taking account of the future rewards in the final state severely impacts performance ontop of the reduction in performance from not utilising the value baseline.

**Figure 3**

In figure 3 we aim to address feedback that we do not clearly show or measure the estimation of the target distribution. We do believe that our performance results are strong evidence for better estimation of the target but agree that some insight into target estimation more directly will help strengthen the conclusions that can be taken away from the paper. Of course in practice the target distribution is unknown. As we do not have perfect advantage estimates. However given a fixed iteration of training, while completely unachievable to perform during training, for the purposes of ablating, we can perform a large scale Monte Carlo Simulation to form unbiased estimates for the advantage function.

In this ablation we perform Monte Carlo estimation of the Q-value using 1280 rollouts to the end of the episode for every state and every action. Advantage estimates can then be formed using the following

$\hat{A}(a_t,s_t) = \hat{Q}(a_t,s_t) - \sum_{a \in A} \pi(a|s_t) \cdot \hat{Q}(a_t,s_t)$.

This allows us to form an estimate of the target distribution for a single state as follows

$q_{i}(a_t|s_t) \propto \pi(a_t|s_t, \theta_i) \exp\left(\frac{\hat{A}(s_t,a_t)}{\eta^*}\right)$.

Using this accurate unbiased estimate we then measure the KL divergence of the Monte Carlo Oracle to SPO which comparatively uses far lower compute. Within this ablation we scale Depth and Number of particles to demonstrate the impact these variables have on estimation. We provide results on Sokoban across 1280 episodes.

The results in Figure 3 demonstrate how important scaling particles is in improving the estimate of the target distribution, with significant drops of KL achieved. We also see that depth contributes to the reduction of the KL divergence. This highlights that methods that estimate the same target but that do not perform planning i.e. using a depth of 0, or limit the number of actions that the distribution is estimated over [1, 69] will have much worse estimation of the target distribution directly impacting the speed of EM training.

---

### Author Response · Authors · 2024-08-12
**Reviewer Engagement**

Dear AC/Reviewers:

We kindly highlight the feedback window is coming to a close, and would appreciate feedback from the reviewers in response to our clarifications of their reviews.

We feel we addressed the raised concerns. Specifically, Reviewer X2nr, we clarified their theoretical concerns and engaged constructively regarding the method description. We had hoped for a further interaction, as per their comment "I’m giving this paper a reject in its current form but I am willing to raise the score if the method or its description is improved.".

Thank you for your time.

---

### Decision · Program_Chairs · 2024-09-25

**Decision:**

Accept (poster)

**Comment:**

The importance of Monte Carlo planning in RL is well known (see AlphaGO), and this paper proposes extensions to Sequential MC. The proposed methods are interesting and justified mathematically. The scores given by the reviewers were mixed, with an average on the positive side. To emphasize the value of the paper, one could mention that the mathematical justification of this paper establishes a close relationship with the use of Monte Carlo methods in Active Inference, which tries to solve reinforcement learning problems without the use of rewards. In particular, the following past NeurIPS paper is very related: “Fountas, Zafeirios, Noor Sajid, Pedro Mediano, and Karl Friston. Deep active inference agents using Monte-Carlo methods. Advances in neural information processing systems 33 (2020): 11662-11675.”. We believe that the current paper could be seen as a step to narrow the gap between traditional reinforcement learning and active inference (which as seen through the work of Fountas et. al is still problematic). Given this parallel interest in MC methods in neuroscience, we believe that the current paper on pushing forward RL and SMC methods is useful, and it may eventually help to bring the two communities together.